# Single cell transcriptomics identifies distinct profiles in pediatric acute respiratory distress syndrome

Tim Flerlage[1,2], Jeremy Chase Crawford [3], E. Kaitlynn Allen [3], Danielle Severns[4], Shaoyuan Tan [1], Sherri Surman[1], Granger Ridout[5], Tanya Novak [6,7], Adrienne Randolph [6,7,8], Alina N. West [2] & Paul G. Thomas [3] ✉

Acute respiratory distress syndrome (ARDS), termed pediatric ARDS (pARDS) in children, is a severe form of acute respiratory failure (ARF). Pathologic immune responses are implicated in pARDS pathogenesis. Here, we present a description of microbial sequencing and single cell gene expression in tracheal aspirates (TAs) obtained longitudinally from infants with ARF. We show reduced interferon stimulated gene (ISG) expression, altered mononuclear phagocyte (MNP) transcriptional programs, and progressive airway neutrophilia associated with unique transcriptional profiles in patients with moderate to severe pARDS compared to those with no or mild pARDS. We additionally show that an innate immune cell product, Folate Receptor 3 (FOLR3), is enriched in moderate or severe pARDS. Our findings demonstrate distinct inflammatory responses in pARDS that are dependent upon etiology and severity and specifically implicate reduced ISG expression, altered macrophage repair-associated transcriptional programs, and accumulation of aged neutrophils in the pathogenesis of moderate to severe pARDS caused by RSV.

Acute respiratory distress syndrome (ARDS; in children termed pARDS for pediatric acute respiratory distress syndrome) is defined by the acute onset of hypoxemic respiratory failure associated with pulmonary infiltrates in the absence of heart failure. Both lung intrinsic (i.e., lower respiratory tract infection (LRTI)) and lung extrinsic (i.e., sepsis) etiologies are recognized causes of ARDS and pARDS; in children, LRTI is the most common cause of pARDS[1].

A dysregulated immune response is thought to contribute to the development of ARDS, which proceeds through three phases[2,3]. The initial stage of ARDS, termed the exudative phase, is mediated by activated innate immune cells, namely resident alveolar macrophages as well as recruited monocytes[4] and neutrophils[3,5–8]. In this stage, inflammatory mediators (cytokines, including TNF, IL-1β, IL-8, and IL-6, acute phase reactants, high mobility group-1 protein, leukotrienes, defensins, and complement) coupled with oxidative stress mediated by free radical production, local activation of the coagulation cascade and endothelial injury, protease production, and surfactant dysfunction have been implicated in preclinical and human studies in mediating lung damage and the ARDS phenotype[3,8]. An important step in the resolution of inflammation is clearance of neutrophils by

[1]Department of Infectious Diseases, St. Jude Children's Research Hospital, Memphis, TN, USA. [2]Division of Pediatric Critical Care Medicine, Department of Pediatrics, University of Tennessee Health Science Center, Memphis, TN, USA. [3]Department of Immunology, St. Jude Children's Research Hospital, Memphis, TN, USA. [4]Department of Pediatrics, University of Tennessee Health Science Center, Memphis, TN, USA. [5]Hartwell Center for Biotechnology, St. Jude Children's Research Hospital, Memphis, TN, USA. [6]Department of Anesthesiology, Critical Care, and Pain Medicine, Boston Children's Hospital, Boston, MA, USA. [7]Department of Anaesthesia, Harvard Medical School, Boston, MA, USA. [8]Department of Pediatrics, Harvard Medical School, Boston, MA, USA. ✉e-mail: paul.thomas@stjude.org

macrophages via efferocytosis, which has previously been shown to be deficient in human patients with ARDS[9]. Following the exudative stage, the proliferative stage involves reconstitution of barrier integrity via provisional matrix elaboration. The final stage is a fibrotic phase, which occurs in a subset of individuals with ARDS and is characterized by exuberant fibrosis that leads to prolonged impairment in gas exchange[2]. To date, the majority of studies characterizing the immunopathogenesis of ARDS have enrolled adult individuals. Thus, less is currently known about pARDS pathogenesis.

Globally, respiratory syncytial virus (RSV) is a significant cause of respiratory tract infections (RTIs) across the age spectrum as well as mortality caused by RTIs in children less than 5 years of age[10]. In a large multicenter study of lower respiratory tract infection (LRTI; pneumonia) leading to hospitalization in the United States, RSV and human rhinovirus (HRV) were the most commonly identified viral etiologies, which together accounted for more than 50% of episodes requiring hospitalization[11]. Nasal bulk transcriptomic studies have previously identified reduced interferon responses[12,13] and lower upper airway viral load[13] in infants with higher disease severity RSV bronchiolitis, compared to those with lower severe disease. Airway neutrophilia is commonly seen in RSV infection in infants, and a dysregulated neutrophil response is thought to play a role in disease severity[14]. Thus, pathologic immune responses, most notably by neutrophils, contribute to RSV disease severity; however, specific mechanistic insight into neutrophil-mediated disease pathogenesis is lacking in general and particularly in the context of pARDS[15].

In this work, we use single cell transcriptomics and ViroCap sequencing on tracheal aspirate (TA) samples to demonstrate etiology-dependent and severity-dependent alterations in airway cell transcriptional responses involving interferon stimulated gene (ISG) expression, macrophage repair programs, and neutrophil ageing in infants with acute respiratory failure (ARF) requiring endotracheal intubation and invasive mechanical ventilation (IMV).

## Results

### Description of patients

We prospectively enrolled 24 (14 male and 10 female) patients zero to six years of age (Supplementary Data 1). Because the TA from patient C6 (six years old, the only enrolled patient greater than 2 years of age) had insufficient cell quantity for the primary study objective, these data are not included in final analyses ($n = 23$ patients). Patients were enrolled into one of three cohorts, including patients with a known or suspected LRTI who met diagnostic criteria for moderate or severe pARDS[16] (P-cohort), patients with known or suspected LRTI who did not meet full diagnostic criteria for pARDS or who developed mild pARDS at any point during follow-up (L-cohort), and patients with ARF without known or suspected LRTI at the time of enrollment (C-cohort; Fig. 1a). Based on the anticipated duration of IMV for each of these cohorts, the study design included collection of up to three TAs from P-cohort patients, up to two TAs from L-cohort patients, and one TA from C-cohort patients. There were no significant differences in timing of sample collection between cohorts at each sampling time point (Fig. 1b).

The 23 patients were zero to two years of age (mean age 5.8 months (mos), standard deviation (SD) 6.2 mos). Twelve patients were enrolled into the P-cohort, 5 into the L-cohort, and 6 into the C-cohort. C7 presented apnea as the primary indication for IMV and was found to have human rhinovirus (HRV) by subsequent testing but was included in the C-cohort to maintain study group allocation based on the data available at enrollment.

We compared illness severity across cohorts using oxygen saturation index (OSI)[17–19] and duration of IMV (Fig. 1c). By OSI, P-cohort patients had significantly higher severity than L- and C-cohort patients. Though there was a difference in mean duration of IMV between P- and L-cohort patients (10.5 vs 7.6 days respectively), this

difference was not statistically significant. Notably, L-cohort patients (mean age 1.2 mos, SD 0.44 mos) were significantly younger than P-cohort (mean age 5.5 months, SD 4.5 months) and C-cohort patients (mean age 10.2 months, SD 9.3 months).

### ViroCap analysis corroborates and supplements clinical microbiological testing

The clinical infectious etiologic diagnosis of ARF for enrolled patients included identification of a viral etiology alone ($n = 9$; Supplementary Data 1 and Fig. S1b), bacterial pathogen alone ($n = 1$), and co-detection ($n = 7$; both viral and bacterial respiratory pathogens identified). As there is no standardized approach to microbiological testing for patients with ARF, we used ViroCap, a viral capture sequencing technique[20] separately on both RNA and DNA extracted from TA supernatant to supplement and further refine clinical diagnoses for patient classification for single cell gene expression (scGEX) analyses based on inciting etiology. In the RNA capture, we identified viral reads in all patients with LRTI as the etiology of respiratory failure (P- and L-cohorts) (Fig. 2a). We were unsuccessful in recovering RNA from any C-cohort TAs, except for HRV-positive C7. The findings from ViroCap sequencing of RNA aligned with clinical testing results, identified additional co-infecting viruses, and suggested an alternative primary etiology of ARF in certain patients.

Influenza A virus (IAV), which is included on the institutional clinical multiplex PCR panel but was detected only in clinical testing obtained from a single patient (P11), was identified in TAs from 5/17 (29%) patients with negative clinical testing for IAV by ViroCap. We further analyzed these data by extracting reads assigned to IAV and mapping these to a reference H3N2 IAV genome using Burrows-Wheeler Aligner (BWA)[21] and VarScan[22] (v2.3.9), which confirmed identification of IAV reads mapping across distinct segments of the influenza genome with patient-specific genetic variation. Although patient P1 was diagnosed with a bacterial LRTI and was negative for HRV via clinical testing, ViroCap identified HRV sequences within the TAs across all three sampling time points. We additionally identified RNA capture reads mapping to primate bocaparvovirus 1 in Samples 1 and 2 from patients P7, P8, P9, P10, P12, L3, and L4, which may indicate concurrent or recent infection with human bocavirus 1 or 3 (these are not included on the institutional respiratory virus detection assay). Finally, despite a clinical diagnosis of adenovirus for patient P4, RNA ViroCap lacked adenovirus reads and instead showed enrichment for reads mapping to anelloviruses (torque teno viruses); only a small number of reads from the DNA capture from the second TA supernatant for this patient matched adenovirus, suggesting a potential disconnect between the positive clinical diagnosis and inferred disease etiology.

In addition to viral respiratory pathogens, we identified previously described constituents of the airway virome, including anelloviruses (AVs); torque teno virus (TTV), torque teno mini virus (TTMV), torque teno midi virus (TTMDV), human herpesviruses (HHVs; HHV1 (herpes simplex virus), 4 (Epstein-Barr virus; EBV), 5 (cytomegalovirus; CMV), 6B (HHV6B), and 7 (HHV7)), human papillomaviruses (HPV), and bacteriophages[23].

We further analyzed the proportion of bacterial reads mapping to *Haemophilus influenzae* (*H. flu*), *Moraxella catarrhalis* (*M. cat*), and *Streptococcus pneumoniae* (*S. pneumo*) in capture sequencing performed on DNA extracted from TA supernatant (Fig. 2b). These bacteria were selected because they were the most common clinically identified bacteria in our cohort and are known to colonize the respiratory tract in infants. Of these bacteria, *M. cat* was the most frequently identified (at least 1 read detected in 35 of 49 TAs (71.4%), including in 3 C-cohort patients). Reads aligning to the *H. flu* genome were detected in 4 patients from whom clinical airway bacterial cultures were not obtained (P006, P010, P011, and L003; Fig. 2b, black asterisk indicates patients for whom no culture was obtained),

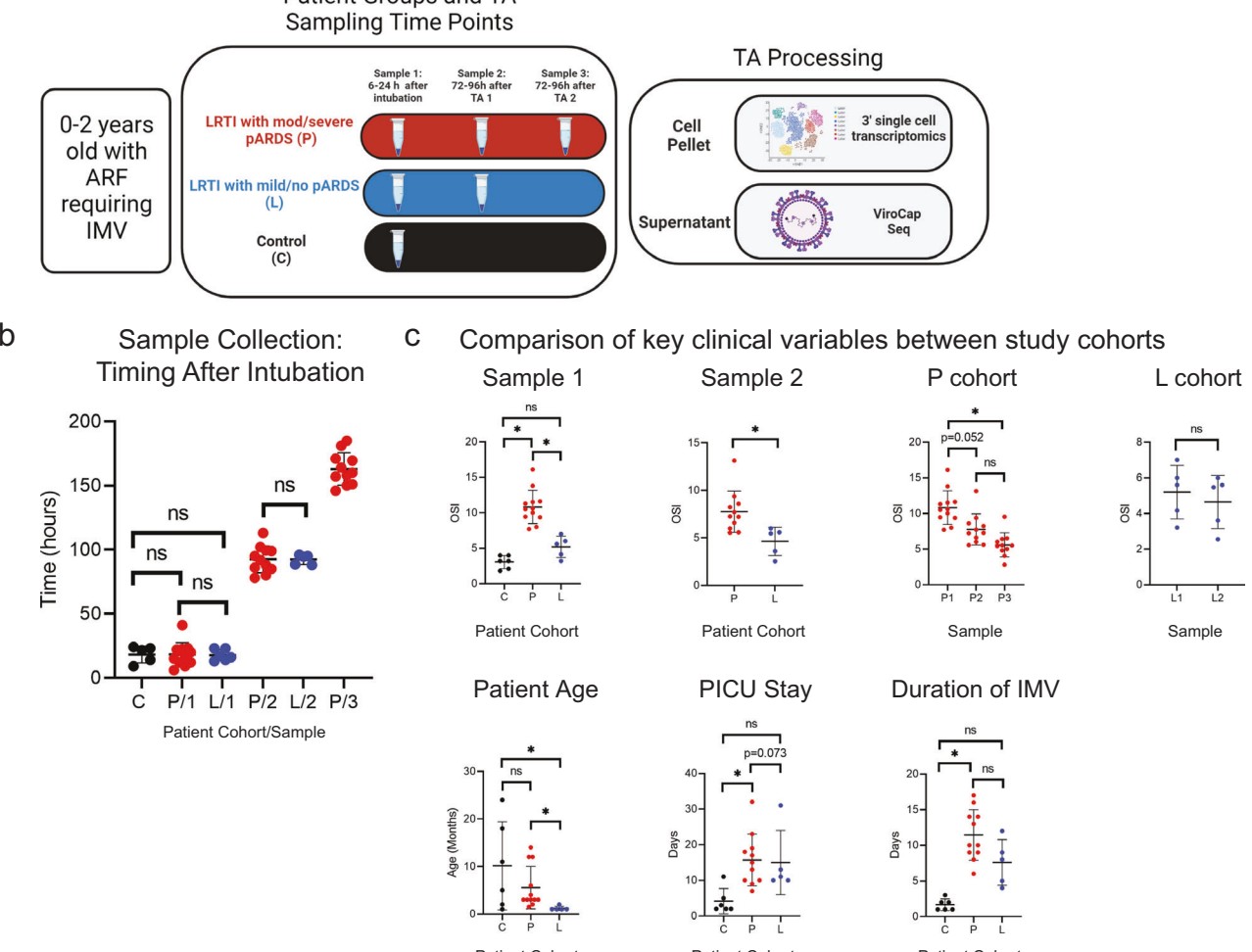

**Fig. 1 | Overview of Study Design and Patient Population. a** Representation of the study population, study design, and primary assays conducted on tracheal aspirate samples. Created with BioRender.com. 3' single cell transcriptomics library preparation was performed using 10X Genomics. **b** Plot of the number of hours elapsed since endotracheal intubation at the time of sample collection for each patient group (*n* = 6 patients for C-cohort, *n* = 12 patients for P/1 and 11 for P/2 and P/3, and *n* = 5 patients for L/1 and L/2). A Kruskal-Wallis test followed by a post-hoc (Dunn's multiple comparison test) was used to test for statistically significant differences between groups at each time point. Error bars represent the mean value for the group +/− SD. Source Data are provided as a Source Data File **C**, Select clinical data abstracted from the electronic medical record (EMR). (*n* = 6 patients for C-cohort, *n* = 12 patients for P-cohort Sample 1 and patient age, *n* = 11 patients for P-cohort Sample 2 and 3 as well as PICU stay and duration of IMV, and *n* = 5

patients for L-cohort). Statistical testing was performed using a two-sided Wilcoxon Rank Sum Test (for statistical testing between two groups) or a Kruskall–Wallis test followed by a post-hoc Dunn's multiple comparison test (for statistical testing between more than two patient groups). Significant differences (*p*-value < 0.05) between groups following post-hoc testing are noted in the plots with an asterisk (Day 1 OSI: P_v_L *p* = 0.041, P_v_C *p* = 0.0002, L_v_C *p* = 0.7367; Day 4 OSI: P_v_L *p* = 0.0053; P-cohort OSI: P1_v_P2 *p* = 0.052, P1_v_P3 *p* < 0.0001, P2_v_P3 *p* = 0.113; L-cohort OSI: L1_v_L2 *p* = 0.5159; Subject age: P_v_L *p* = 0.0176, P_v_C *p* > 0.999, L_v_C *p* = 0.02; PICU LOS P_v_L *p* > 0.999, P_v_C *p* = 0.0082, L_v_C *p* = 0.073; Duration IMV: P_v_L *p* = 0.469, P_v_C *p* = 0.0004, L_v_C *p* = 0.157). Error bars represent the mean value for the group +/− SD. Source Data are provided as a Source Data File. OSI Oxygen Saturation Index, IMV Invasive Mechanical Ventilation.

---

suggesting that these patients may have had both RSV and *H. flu* present in high proportional abundance at the onset of ARF. In response to antibiotic administration, the abundance of *H. flu* reads decreased in TA samples; however, similar decreases were not witnessed in the frequency of *M. cat* reads (Fig. 2b, black "a" indicates administration of antibiotics).

Based on clinical diagnoses and results of ViroCap sequencing, which identified RSV as the inciting etiology of ARF in the majority of P-cohort (7/12, 58.3%) and all L-cohort patients, we further divided P-cohort patients into RSVneg (*n* = 5) and RSVposP (*n* = 7) groups for comparison with L-cohort patients (RSVposL) and C-cohort patients (Con) (Figs. 2c and S1b) in our scGEX analyses. We noted differences in OSI and ventilator support requirements in Samples 1 and 2 between

groups and there was not a statistically significant difference between the ages of patients in the RSVposP and RSVposL groups (Figs. 2d and S1a).

## Single cell transcriptomics identifies differences in TA cellular composition and transcriptional state that depend on etiology, illness severity, and time after intubation

We present transcriptional data from 84,495 cells obtained from 51 TA samples (35 from P-cohort, 10 from L-cohort, and 6 from C-cohort patients). The initial unsupervised clustering analysis identified 15 transcriptionally distinct cell clusters (Fig. S2a), which were further grouped by gene expression into 11 distinct cell populations (Fig. 3a). In this process, we separated putative monocyte populations (*FCGR3B*

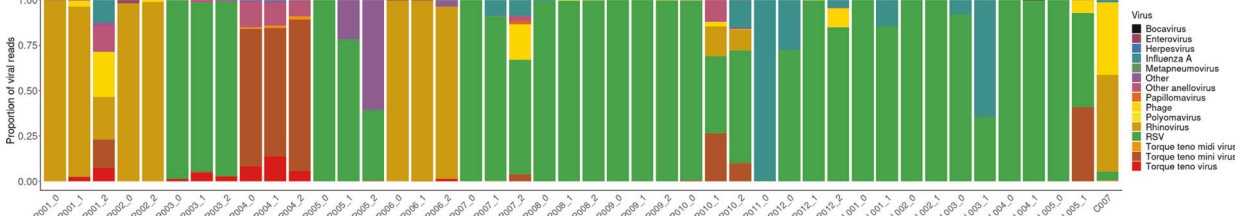

a Proportion of viral reads mapping to an individual virus

b Proportion of reads mapping to *H. flu*, *M. cat*, or *S. pneumoniae*

c Patient group allocation for scGEX analyses based on ViroCap results

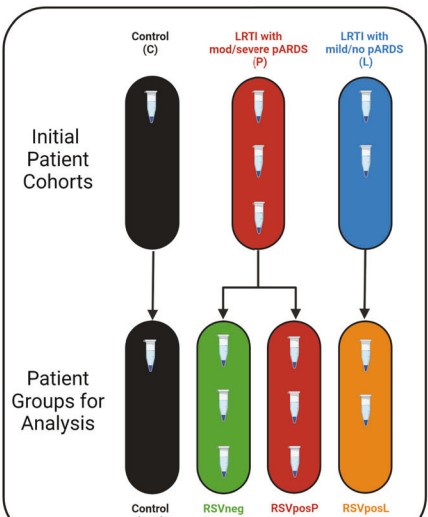

d Comparison of key clinical variables between patient groups for scGEX analyses

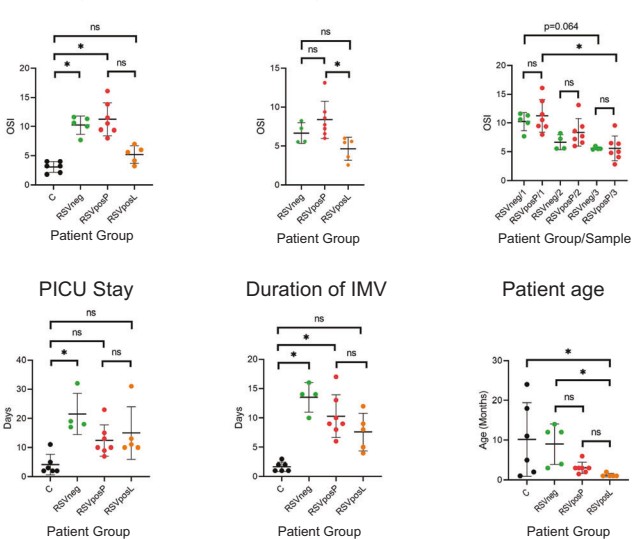

**Fig. 2 | ViroCap sequencing in tracheal aspirate supernatant confirms and supplements clinical microbiology testing. a** Proportion of total viral reads mapping to individual viruses in ViroCap sequencing performed on RNA extracted from TA supernatant. Source Data are provided as a Source Data File. **b** Proportion of total bacterial reads mapping to *Haemophilus influenzae* (*H. flu*), *Moraxella catarrhalis* (*M. cat*), and *Streptococcus pneumoniae* (*S. pneumoniae*) in ViroCap sequencing performed on DNA extracted from TA supernatant. Black "a" in bars indicate that the patient was receiving antibiotics at the time of sample collection. Black asterisk below the x-axis indicate that no bacterial culture was obtained from the patient for routine clinical testing. Source Data are provided as a Source Data File. **c** Schematic demonstrating how Patient groups were created for single cell gene expression (scGEX) analyses from initially enrolled cohorts. Created with BioRender.com. **d** Select clinical data abstracted from the EMR for each Patient group (*n* = 6 patients for C group, *n* = 5 patients for RSVneg group Sample 1 and patient age, *n* = 4 patients for RSVneg group Sample 2 and 3 as well as PICU stay and duration of IMV, *n* = 7 patients for RSVposP, and *n* = 5 patients for RSVposL). Statistical testing was performed between groups using a Kruskall-Wallis test followed by a post-hoc Dunn's multiple comparison test. Significant differences between

groups following post-hoc testing are noted with an asterisk (Sample 1: C_v_RSVneg *p* = 0.0068, C_v_RSVpos *p* = 0.0014, C_v_RSVposL *p* > 0.999, C_v_RSVposL *p* > 0.999, RSVneg_v_RSVposL *p* = 0.269, RSVposP_v_RSVposL *p* = 0.13; Sample 2: RSVneg_v_RSVposP *p* = 0.699, RSVneg_v_RSVposL *p* = 0.454, RSVposP_v_RSVposL *p* = 0.011; P-cohort OSI: RSVneg1_v_RSVposP1 *p* > 0.999, RSVneg2_v_RSVposP2 *p* > 0.999, RSVneg3_v_RSVposP3 *p* > 0.999, RSVneg1_v_RSVneg2 *p* = 0.788, RSVneg1_v_RSVneg3 *p* = 0.064, RSVposP1_v_RSVposP2 *p* > 0.999, RSVposP1_v_RSVposP3 *p* > 0.0073, RSVposP2_v_RSVposP3 *p* = 0.775, RSVneg2_v_RSVneg3 *p* > 0.999; PICU stay: C_v_RSVneg *p* = 0.0033, C_v_RSVposP *p* = 0.271, C_v_RSVposL *p* = 0.145, RSVneg_v_RSVposP *p* = 0.4499, RSVneg_v_RSVposL *p* > 0.999, RSVposP_v_RSVposL *p* > 0.999; Duration of IMV: C_v_RSVneg *p* = 0.0014, C_v_RSVposP *p* = 0.0144, C_v_RSVposL *p* = 0.315, RSVneg_v_RSVposP *p* > 0.999, RSVneg_v_RSVposL *p* = 0.439, RSVposP_v_RSVposL *p* > 0.999; Patient Age: C_v_RSVneg *p* > 0.999, C_v_RSVposP *p* > 0.999, C_v_RSVposL *p* = 0.041, RSVneg_v_RSVposP *p* = 0.716, RSVneg_v_RSVposL *p* = 0.0095, RSVposP_v_RSVposL *p* = 0.381). Error bars represent the mean value for the group +/− SD. Source Data are provided as a Source Data File. OSI Oxygen Saturation Index, IMV Invasive Mechanical Ventilation, PICU Pediatric Intensive Care Unit.

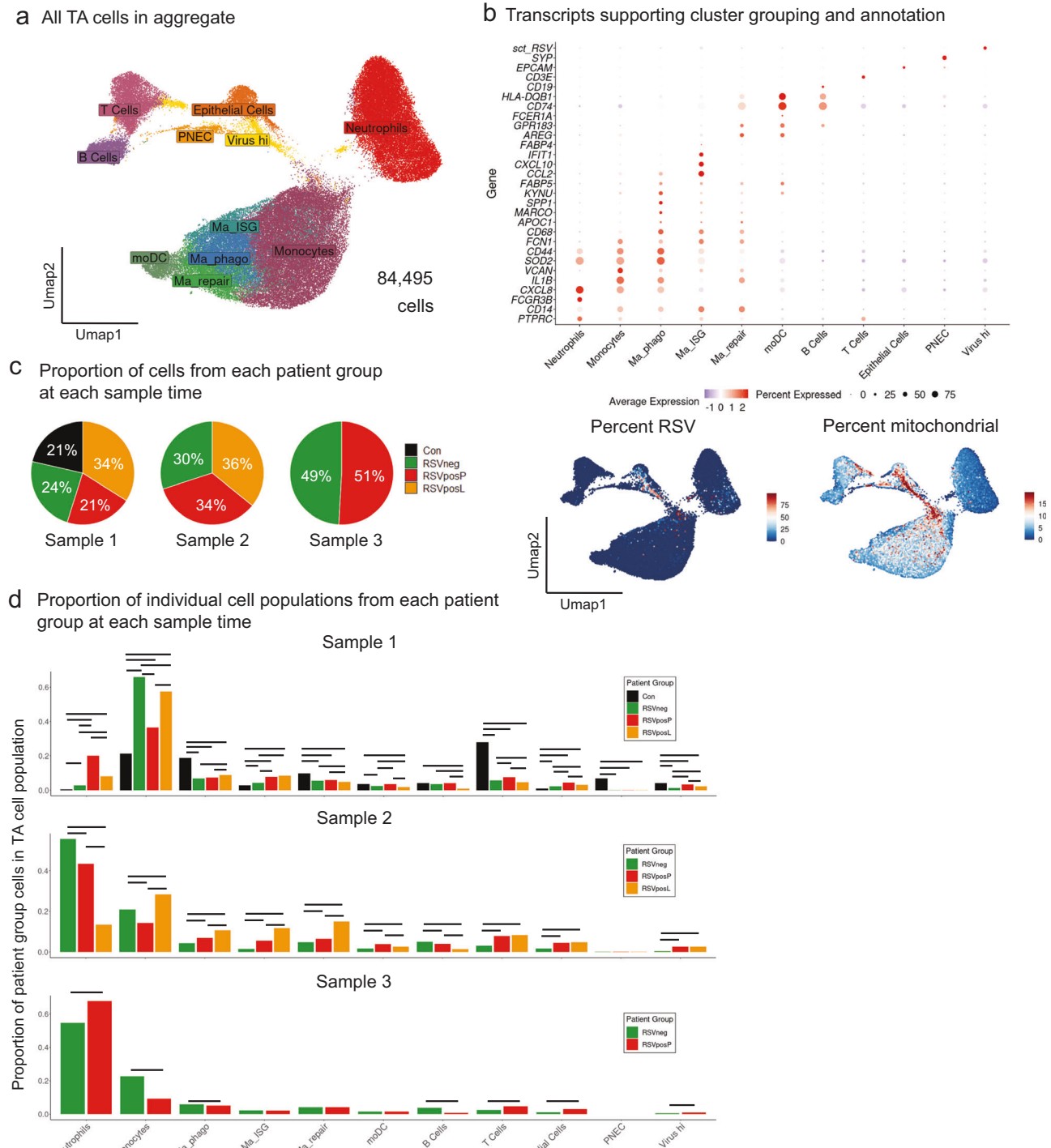

**Fig. 3 | Single cell transcriptomics identifies illness severity and etiology-dependent differences in gene expression. a** Uniform manifold approximation and projection (UMAP) of all cells included in the final data set. **b** (Top) Dot plot of differentially expressed genes that identify individual cell populations as labeled in Fig. 3a. The size of the dot indicates percent of cells within an individual cluster expressing a particular transcript, and the color indicates the average expression of the transcript within the cell population. (Bottom) Feature plots (UMAP as in Fig. 3a) demonstrating percent RSV (left) and percent mitochondrial (right) gene expression. **c** Pie charts demonstrating percentages of cells from each patient group in aggregate single cell data at each Sample timepoint. Source Data are provided as a Source Data File. **d** Bar plot of proportional abundance of cells from each Patient group within each cell population as labeled in Fig. 3a at each Sample timepoint. Statistical testing was performed using a pairwise comparison of proportions with *p*-value adjustment using the Bonferroni correction. Lines above bars in the plot indicate significant differences in proportions between groups. A significance threshold of $p_{adj} < 0.05$ was used to identify statistical differences. Source Data are provided as a Source Data File.

low, *CD14* pos, *VCAN* hi) from neutrophils (*FCGR3B* hi, *CD14* low, *VCAN* low) and macrophages (*CD68*, *MARCO*, and *APOC1* pos). Within macrophage cell clusters, we further divided cells into putative functional phenotypes, including phagocytic macrophages (Ma_phago), which demonstrated increased expression of genes associated with phagocytosis and respiratory burst (*SOD2* and *CD44*), interferon stimulated gene (ISG) expressing macrophages (Ma_ISG), which upregulated ISGs and cytokines (*IFIT1*, *CXCL10*), and repair-associated macrophages (Ma_repair), which differentially upregulated genes associated with repair (*AREG*, *IL10*) and antigen presentation/adaptive immune cell interaction (*CD74*, major histocompatibility type 2 (MHC II) genes, and *GPR138*; Fig. 3a,b).

Based on clinical and ViroCap data (Fig. 2a), we included the genomes for RSV, enteroviruses B and D, HRV A, B, and C, torque teno virus (TTV), torque teno mini virus (TTMV), IAV, and adenoviruses B and C in a custom reference used to align our single-cell RNA sequencing data. Of the viruses included in the custom reference, we identified reads aligning to RSV and, to a lesser extent, IAV genomes (Fig. S2b). A subset of cells within the epithelial compartment had particularly high expression of transcripts mapping to the RSV genome (Fig. 3b, bottom left) and were analyzed separately as Virus hi cells. This population demonstrated elevated expression of mitochondrial transcripts (Fig. 3b, bottom right) relative to other cell populations, suggesting that they were undergoing lysis or cell death.

After annotating cell populations by lineage and function in the aggregate data set (Fig. 3a), we next began to approach the primary hypotheses of the study by comparing differences in proportional abundance and transcriptional profile of these cell populations by illness severity, inciting etiology of ARF, and time after onset of ARF. Though there was a higher percentage of Sample 1 cells obtained from RSVposL group TAs, proportions of cells from each subgroup were similar at Sample 2 and 3 timepoints (Fig. 3c). The most notable differences between patient groups were in innate immune cell populations in Samples 1 and 2. While RSVneg and RSVposP TAs were significantly enriched in neutrophils, RSVposL TAs were found to have an increase in monocytes and macrophages (Figs. 3d and S2c).

Using CellChat[24], an algorithm used to measure ligand-receptor interactions between cells, we found that, in general, cellular interactions were fewer and weaker between cell populations in RSVposP TAs compared to RSVposL and RSVneg TAs. Specifically, Ma_repair and Ma_ISG macrophage populations are major sources of increased outgoing signaling strength in RSVposL TAs compared to RSVposP TAs (Fig. S3d, top) and monocytes, along with Ma_repair populations, are major sources of outgoing signaling strength in RSVneg TAs compared to RSVposP TAs (Fig. S3d, bottom).

Together, these data demonstrate that the development of airway neutrophilia, reduced interferon responsiveness, and reduced ligand-receptor interactions between MNPs and other TA cellular populations communication are key discriminatory features of moderate to severe RSV-associated pARDS compared to milder RSV-associated ARF phenotypes.

### Epithelial responses to LRTI-associated ARF depend on primary etiology and illness severity

Respiratory viruses, including RSV, IAV, and HRV, display tropism for epithelial cells lining the airway, from the nasopharynx to the alveoli. Thus, epithelial cell responses to these viral pathogens are of critical importance. We therefore subclustered non-pulmonary neuroendocrine cell (PNEC) populations expressing *EPCAM* (clusters 11 and 14) from the aggregate data. Among these 3332 epithelial cells, 11 transcriptional states were detected using unsupervised clustering analysis (Fig. 4a). We identified transcriptional states representative of intermediates along the epithelial repair trajectory[25], including basal cells (*KRT5*), secretory cells (*MUCSAC*, *SCGB1A1*), deuterosomal cells (*HES6*), *FOXN4* positive cells[26], and ciliated epithelial cells (*FOXJ1*, *PIFO*).

Additionally, there was a population of cells likely originating from tracheal hillocks[27] in light of differential upregulation of *KRT13* and *CXCL17* (Fig. 4b). We also identified distinct clusters of transcriptional states (Fig. 4b), which were composed partially of cells expressing innate immune transcripts (*PTPRC*, *FCGR3B*, *VCAN*). We speculate that these clusters represent immune cells interacting with epithelial cells at the time of cell capture; however, we cannot exclude that these represent technical doublets.

Given the enrichment for RSV LRTI in P- and L-cohorts (Figs. 2a and S1b), we focused further analyses of epithelial cell populations on RSV-host interactions and on comparisons between patient groups. We subsetted cells with any RSV expression (RSV-infected) from those with no RSV infection (bystander). The majority of RSV-infected cells (59.8%) were from RSVposL patients, who had a significantly higher proportion of RSV-infected cells amongst all epithelial cells (Fig. 4c). Interestingly, a small percentage of RSV-infected epithelial cells were from RSV-neg patients (5.9%) and C-cohort patients (1.9%). As persistent RSV infection has been identified previously[28], these findings may indicate recent RSV infections.

We next analyzed normalized *RSV* expression levels within epithelial cells from each comparator group, which demonstrated significantly increased *RSV* expression levels in RSVposL cells (Fig. 4c, middle top). However, there was no significant difference in normalized *RSV* expression when comparing subsetted RSV-infected epithelial cells from RSVposL and RSVposP groups (Fig. 4c, middle bottom). We then analyzed percent mitochondrial and *RSV* gene expression within the subsetted RSV-infected epithelial cells and found that while there was no significant statistical difference between percent of the cellular transcriptome occupied by RSV between RSVposP and RSVposL groups (Fig. 4c, right), there was significantly less mitochondrial gene expression in RSVposL groups compared to all other patient groups (Fig. 4c, right). Consistent with these findings, our ViroCap data demonstrated a non-significant trend towards higher RSV counts in RSVposL TA supernatants compared to RSVposP TA supernatants (Fig. 4d).

We identified significantly increased ISG module[29] usage by subsetted epithelial cells from RSVposL patients compared to the other three groups (Fig. 4e). Using pathway enrichment analysis (MSigDB Hallmark 2020 Database) to compare RSVposL and RSVposP epithelial cells, we found that inflammatory, interferon, and epithelial mesenchymal transition pathways were enriched in RSVposL relative to RSVposP epithelial cells (Fig. 4f). Using DGE testing, generated a list of the top 10 transcripts differentially upregulated (by average $\log_2$ fold change) by P and L cohort epithelial cells for module scoring (Supplementary Data 2). We then used these module scores to assess their ability to differentiate P versus L cohort patient origin amongst epithelial cells (Fig. 4g) using receiver operating characteristic (ROC) curves. The area under the ROC curve score (AUROC) score for the P cohort epithelial module was 0.7 and the AUROC score for the L cohort epithelial module was 0.78.

### MNP transcriptional response and developmental trajectory is etiology- and severity dependent

Mononuclear phagocytes (MNPs), including macrophages and monocytes, have been implicated in the continuum of host response to pARDS, from initiation to recovery[2]. To study MNP responses in greater detail, we subclustered MNPs from the aggregate data set (Fig. 2a), which included 47,266 cells grouped into 14 clusters by unsupervised clustering analysis (Fig. 5a). We used the transcripts *FCN1* (inflammatory monocyte-like phenotype), *SPP1* (repair type phenotype), and *FABP4* (resident macrophage phenotype)[30,31], along with *CXCL10* (interferon stimulated macrophages), *FCER1A* (dendritic cells), and *IL6* (a *FCN1* low, SPP1 low cytokine producing macrophage population) to annotate subclustered MNPs. We identified regional upregulation of these transcripts within MNP gene

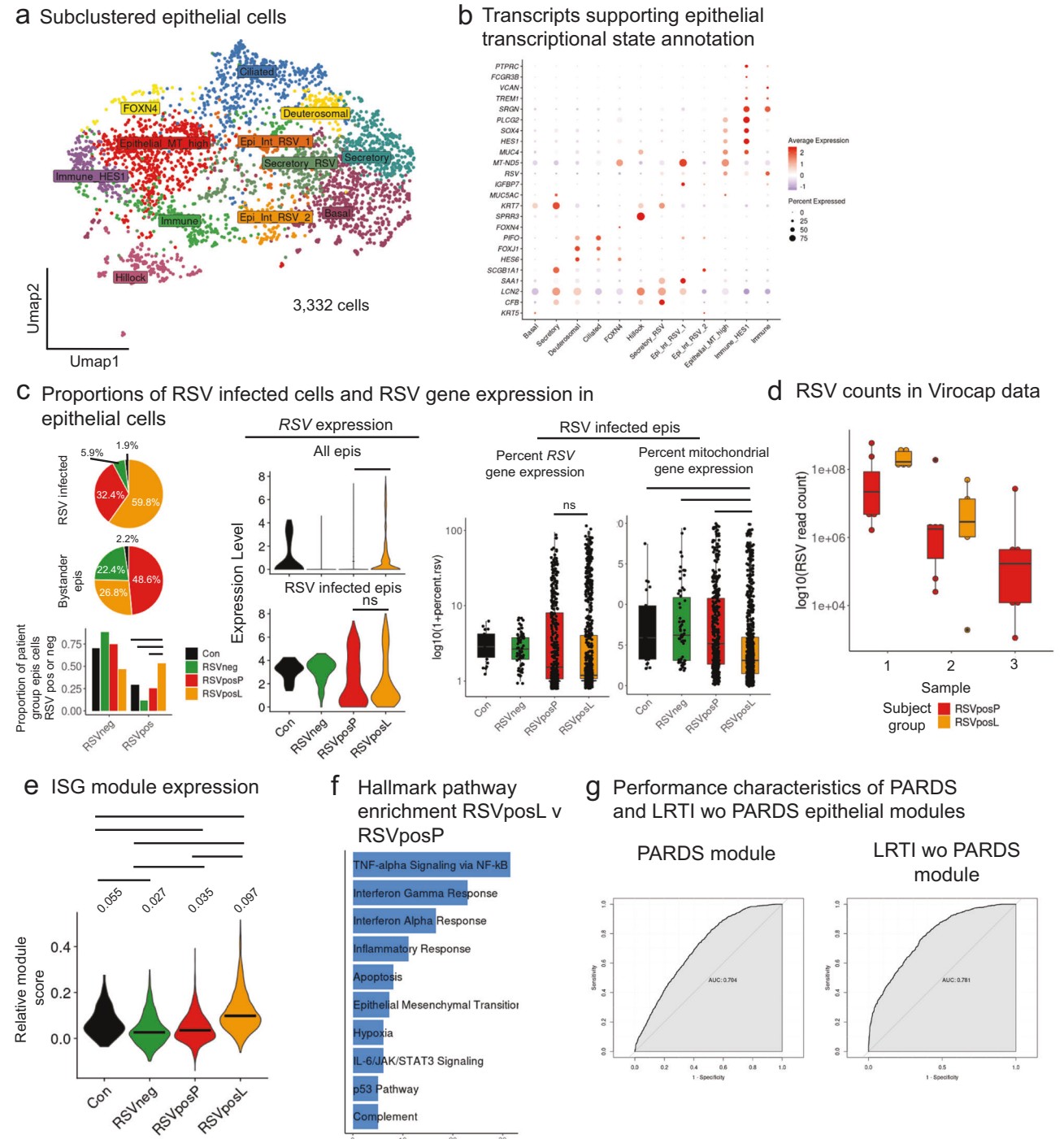

**a** Subclustered epithelial cells

**b** Transcripts supporting epithelial transcriptional state annotation

**c** Proportions of RSV infected cells and RSV gene expression in epithelial cells

**d** RSV counts in Virocap data

**e** ISG module expression

**f** Hallmark pathway enrichment RSVposL v RSVposP

**g** Performance characteristics of PARDS and LRTI wo PARDS epithelial modules

expression space, which was largely represented by *FCN* high inflammatory populations (Figs. 5a, b, and S3a). Though isolated from a more proximal airway compartment, these populations are reminiscent of those described in bronchoalveolar lavage fluid (BALF) from patients with COVID-19[30].

Given similar MNP gene expression profiles, we used gene lists generated by Liao et al.[30] along with those described by Reyes et al.[32,33] to perform module scoring (Fig. S3b, c). We found that the gene module groups described by Liao et al. discretely localized within MNP gene expression space. Group 1 gene module usage was highest in *FCN1* high clusters and was significantly increased in RSVneg MNPs compared to other groups, group 1_2 module (enriched in ISGs) usage localized within areas of high *CXCL10* and ISG module[29] usage (Fig. S3b) and was significantly increased in RSVposL MNPs, and group

2 module usage was enriched in *SPP1* high clusters and was also significantly increased in RSVposL MNPs. Together, these data suggest that airway MNP responses are relatively conserved amongst patients with virus induced ARF.

Consistent with previous analyses of proportional abundance of cell phenotypes within patient groups (Fig. 3d), a significantly higher proportion of Samples 1 and 2 MNPs were obtained from RSVposL patients (Fig. 5c). We analyzed the proportional abundance of MNPs within each cluster by patient group and Sample time point (Fig. S3d) and noted a higher abundance of RSVposL and RSVposP MNPs in cluster 7 (corresponding to *CXCL10*/ISG module high MNPs), RSVneg MNPs in clusters 2, 3, 10 (localized in *FCN1* high *SPP1* neg gene expression space), RSVposL MNPs in clusters 0, 4, and 11 (most notably from Sample 1 to Sample 2).

**Fig. 4 | Reduced ISG and cytokine signaling in epithelial cells from patients with moderate to severe pARDS. a** UMAP of epithelial cell populations subclustered from the aggregate data set and labeled by putative cell type based on gene expression. **b** Dot plot of differentially expressed genes that define individual cell populations within subclustered epithelial cells. Cell populations are labeled as in Fig. 4a. The size of the dot indicates percent of cells within an individual cluster expressing a particular transcript, and the color indicates the average expression of the transcript within the cell population. **c** (Left) Pie charts of the proportion of RSV infected and bystander epithelial cells from each patient group (Left, top) and bar plot of the proportion of all epithelial cells from each group that are RSV-infected or bystander populations (Left, bottom; $n = 1125$ RSV infected cells, $n = 2207$ bystander cells, and $n = 3332$ total epithelial cells). Statistical testing was performed using pairwise comparison of proportions with $p$-value adjustment for multiple comparisons using the Bonferroni correction. Lines above bars in the plot indicate when a significant difference in proportions between groups was found using a significance threshold of $p_{adj} < 0.05$; (Middle) Violin plots of RSV transcript expression levels within all epithelial cells (Middle, top) and within subsetted RSV-infected epithelial cells (Middle, bottom); (Right) Box plots of percentage of the cellular transcriptome occupied by RSV within RSV infected epithelial cells and percentage of the cellular transcriptome occupied by mitochondrial transcripts within RSV-infected epithelial cells. Testing for statistically significant differences between groups was performed using two-sided pairwise Wilcoxon rank sum testing with $p$-value adjustment using the Bonferroni correction. Significant differences (at a threshold of $p_{adj} < 0.05$) between groups are denoted with a bar and labeled as not significant ("ns") as appropriate for comparisons of primary interest (All epis RSV expression: RSVposL_v_RSVposP $p_adj = 9.9e\text{-}34$; RSV infected epis RSV expression: RSVposL_v_RSVposP $p_adj = 1$; Percent mitochondrial: RSVpos_v_C $p_adj = 0.015$,

RSVposL_v_RSVneg $p_adj = 3.2e\text{-}7$, RSVposLvRSVposP $p_adj = 1.2e\text{-}13$; Percent RSV: RSVposL_v_RSVposP $p = 0.369$). The upper whisker extends from the hinge to the highest value that is within 1.5*IQR of the hinge, where IQR is the inter-quartile range, or distance between the first and third quartiles. The lower whisker extends from the hinge to the lowest value within 1.5*IQR of the hinge. Data beyond the end of the whiskers are outliers and plotted as points (as specified by Tukey). Source Data are provided as a Source Data File. **d** Box plot of RSV read counts in RSVposP and RSVposL TA supernatant ($n = 7$ patients for RSVposP and $n = 5$ patients for RSVposL). The upper whisker extends from the hinge to the highest value that is within 1.5*IQR of the hinge, where IQR is the inter-quartile range, or distance between the first and third quartiles. The lower whisker extends from the hinge to the lowest value within 1.5*IQR of the hinge. Data beyond the end of the whiskers are outliers and plotted as points (as specified by Tukey). Source Data are provided as a Source Data File. **e** Violin plot of ISG[29] (with the addition of *IFI6*) module usage by all epithelial cells from each patient group. Testing for differences between groups was performed using a Kruskal Wallis Test as well as a pairwise Wilcoxon Rank Sum Test with $p$-value adjustment using the Benjamini & Hochberg approach. Significant differences between groups ($p_{adj} < 0.05$) are denoted using lines and median module usage is denoted with a bar in the violin and further annotated numerically above the violin. Source Data are provided as a Source Data File. **f** Bar plot of Hallmark gene set enrichment analysis (MSigDB Hallmark 2020) for pathways enriched in RSVposL compared to RSVposP epithelial cells. Differentially regulated pathways were identified using a two-sided Wilcoxon Rank Sum Test. **g** Receiver Operator Characteristic (ROC) curves for P and L cohort epithelial gene module differentiating between P and L cohort epithelial cells. AUC denotes Area Under the ROC curve. Source Data are provided as a Source Data File.

Using differential gene expression (DGE) we identified defining characteristics of MNP transcriptional phenotypes enriched amongst patient groups (Supplementary Data 2). Cluster 11 macrophages were transcriptionally distinct from the previously described *SPP1*-high reparative macrophage populations[31] and expressed *IL10* (Fig. 5b) and the scavenger receptor *OLR1* (Fig. 5d), which suggests a particular role for this population in immune regulation and efferocytosis[34]. Cluster 1 MNPs (Neutrophil-like) demonstrated higher expression of transcripts more commonly associated with neutrophils (*CD177, LCN2, PI3, CD24*, among others; Fig. 5b). Clusters 2, 3, and 10 were most consistent with an inflammatory monocytic phenotype (expression of S100 transcripts, *VCAN*, and *RETN*), which aligns with usage of gene modules from previously published data sets (Fig. S3b). Using the top 20 (by average $\log_2$ fold change) differentially upregulated transcripts, we confirmed that these MNP transcriptional states were enriched in patient groups (i.e., cluster 2/3/10 gene module usage was enriched in RSVneg MNPs) (Figs. 5d and S3e and Supplementary Data 2) and observed spatial localization of MNP derived gene set usage within subclustered MNPs (Fig. 5e).

Finally, using pathway enrichment analysis (MSigDB Hallmark 2020 Database; Fig. 5f), we identified enrichment of inflammatory response and interferon signaling pathways amongst RSVposL MNPs compared to RSVposP MNPs, which is consistent with our interpretation of comparisons of module usage between groups. Taken together, these data identify altered macrophage developmental trajectories that depend on the illness severity and underlying etiology of ARF and specifically demonstrate deficient MNP interferon responses as well development of *IL10* expressing repair-associated macrophages in patients with moderate to severe pARDS compared to those with no or mild RSV-induced pARDS.

### Specific aged neutrophil responses in moderate to severe RSV-associated pARDS

As airway neutrophilia has been implicated in the pathogenesis of pARDS and severe RSV infection, we analyzed transcriptional responses in neutrophil populations from the aggregate dataset (Fig. 3a) on the basis of *FCGR3B* expression, which included 23,184 cells. We observed a gradient of *CXCR2* and *CXCR4* (Fig. 6a), consistent with

differences in neutrophil maturity, from mature to aged. Using DGE analyses, we further defined these neutrophil subpopulations and identified increased expression of transcripts encoding S100 proteins, adhesion proteins (*VNN2, SELL, GCA*), and *FOLR3* in mature neutrophils and increased expression of cytokines (*CCL3L1, CCL4L2*), ISGs (i.e., *IFI30*), and transcription factors (i.e., *HES4*) in aging *CXCR4*-pos neutrophils. The population that we termed intermediate, had higher average expression of IL1 transcripts (*IL1RN* and *IL1B*) as well as transcripts that mediate cell survival (*TNFAIP3* and *IER3*; Fig. 6b).

We used previously published[29] neutrophil gene sets to perform module scoring to define putative function and responses within subsetted neutrophils (Fig. S4a). Usage of each of these modules was significantly higher in Sample 1 RSVposP neutrophils compared to other patient groups (Fig. S4a, top), but was significantly lower than RSVposL neutrophils at the Sample 2 time point. Module usage for each gene set was localized in gene expression space to mature and intermediate neutrophils, and less so to aged neutrophils. These data provided the initial suggestion of longitudinal differences in neutrophil transcriptional responses in RSVposP patients.

Consistent with previous analyses of proportional abundance of cell phenotypes (Fig. 3d), there was an increase in the proportion of total neutrophils from patients with moderate to severe pARDS (most notably from RSVneg patients) between Samples 1 and 2 (Fig. S4b). Within neutrophil phenotypes, we identified a significant increase in the proportional abundance of RSVposP neutrophils consistent with an aged transcriptional phenotype over the course of the sampling time points (Fig. 6c), which was not observed in RSVposL or RSVneg neutrophils over the same time points.

To better characterize differences in aged neutrophil gene expression in RSVposP TAs, we identified transcripts differentially upregulated by aged neutrophils between patient groups (Figs. 6d and S4c and Supplementary Data 2). In this analysis we identified differentially increased expression of protease inhibitors (*PI3, SLPI*, and *CSTB*) and *HLA-C* (type I major histocompatibility complex) and differentially decreased expression of cytokines (*CCL3L1* and *CCL4L2*) and mitochondrial genes (*MT-CO1*) in aged RSVposP neutrophils relative to other patient groups. Together, the pattern of differentially expressed transcripts suggested an aged neutrophil response specific to RSVposP

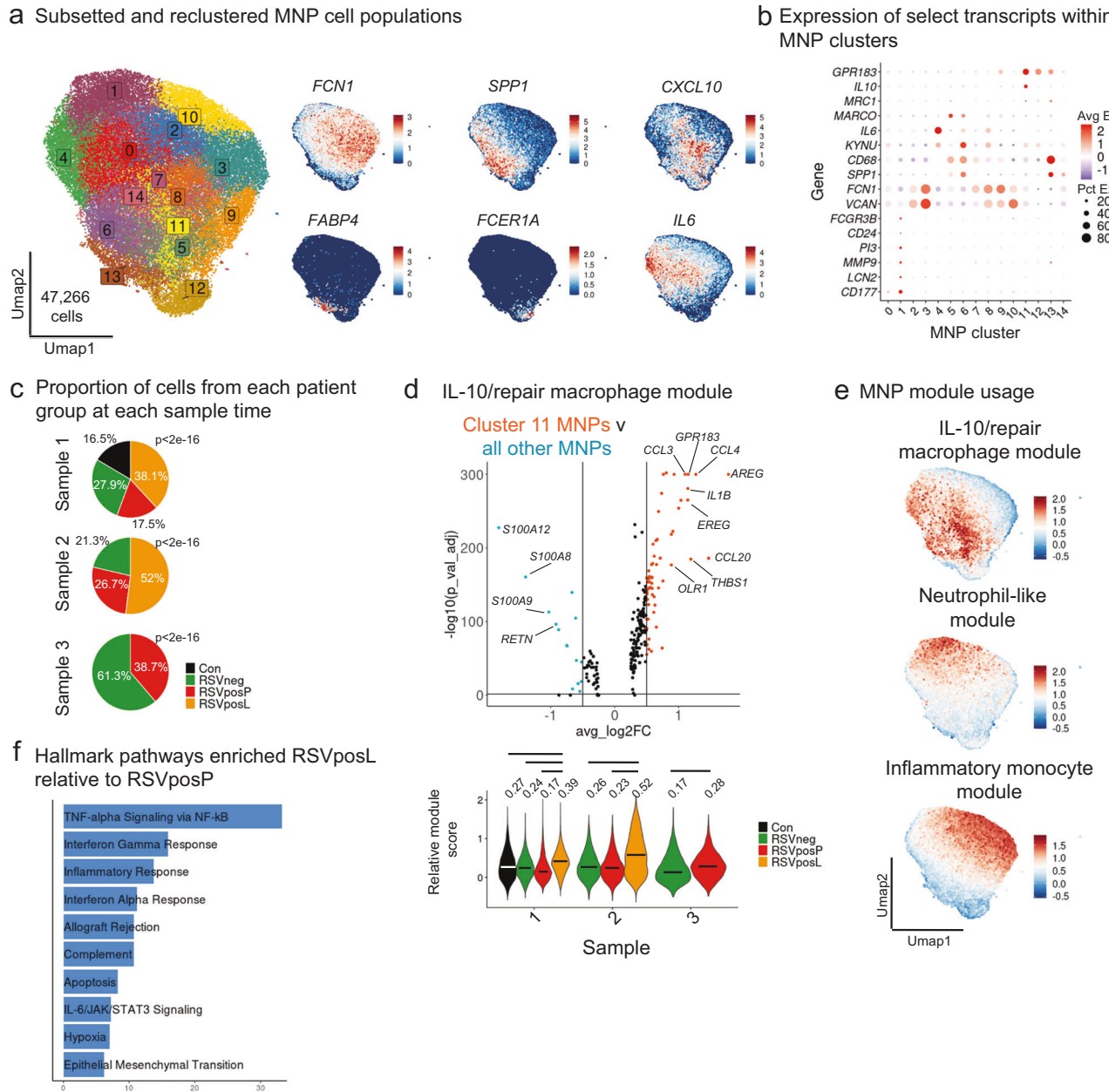

**Fig. 5 | Identification of distinct etiology- and illness severity-specific MNP transcriptional programs. a** (Left) UMAP of subclustered MNPs from aggregate dataset, labeled by cluster number. (Right) Feature plots demonstrating expression levels of select transcripts within gene expression space, as indicated over each UMAP. **b** Dot plot of the expression level of select differentially expressed transcripts within MNP clusters. The size of the dot indicates percent of cells within an individual cluster expressing a particular transcript, and the color indicates the average expression of the transcript within the cell population. Clusters are labeled as in Fig. 5a. **c** Pie charts demonstrating the proportion of MNPs from each patient group at each Sample time point. Testing for significant differences between groups was performed using pairwise comparison of proportions with p-value adjustment using the Bonferroni correction (more than 2 groups being compared) and a chi-squared test with Yates continuity correction and p-value adjustment using the Bonferroni correction (two groups being compared). Source Data are provided as a Source Data File. **d** (Top) Volcano plot demonstrating DEGs between MNP cluster 11 and all other MNPs. DGE testing was performed using Seurat's

FindAllMarkers in default settings (Wilcoxon rank sum test with Bonferroni p-value adjustment) comparing cluster 11 to all other MNPs (a positive avg_log2FC indicates increased expression by cluster 11, and a negative avg_log2FC indicates increased expression by all other MNPs). (Bottom) Violin plot of cluster 11 module usage, grouped by Sample time point and split by patient group. Testing for statistical differences included Kruskal-Wallis and two-sided pairwise Wilcoxon rank sum testing followed by p-value adjustment using the Benjamini & Hochberg method (for comparisons between more than 2 groups) and an unpaired two-sided t test (for comparisons between two groups). Significant differences (below a threshold of $p_{adj} < 0.05$) between groups are indicated with a line and the median module usage for each group and sample is denoted with a bar in the violin and further annotated numerically above the violin. Source Data are provided as a Source Data File. **e** Feature plots demonstrating the usage of novel MNP gene modules in MNPs. **f** Bar plot of Hallmark gene set enrichment analysis (MSigDB Hallmark 2020) for pathways enriched in RSVposL compared to RSVposP MNPs. Differentially regulated pathways were identified using a two-sided Wilcoxon Rank Sum Test.

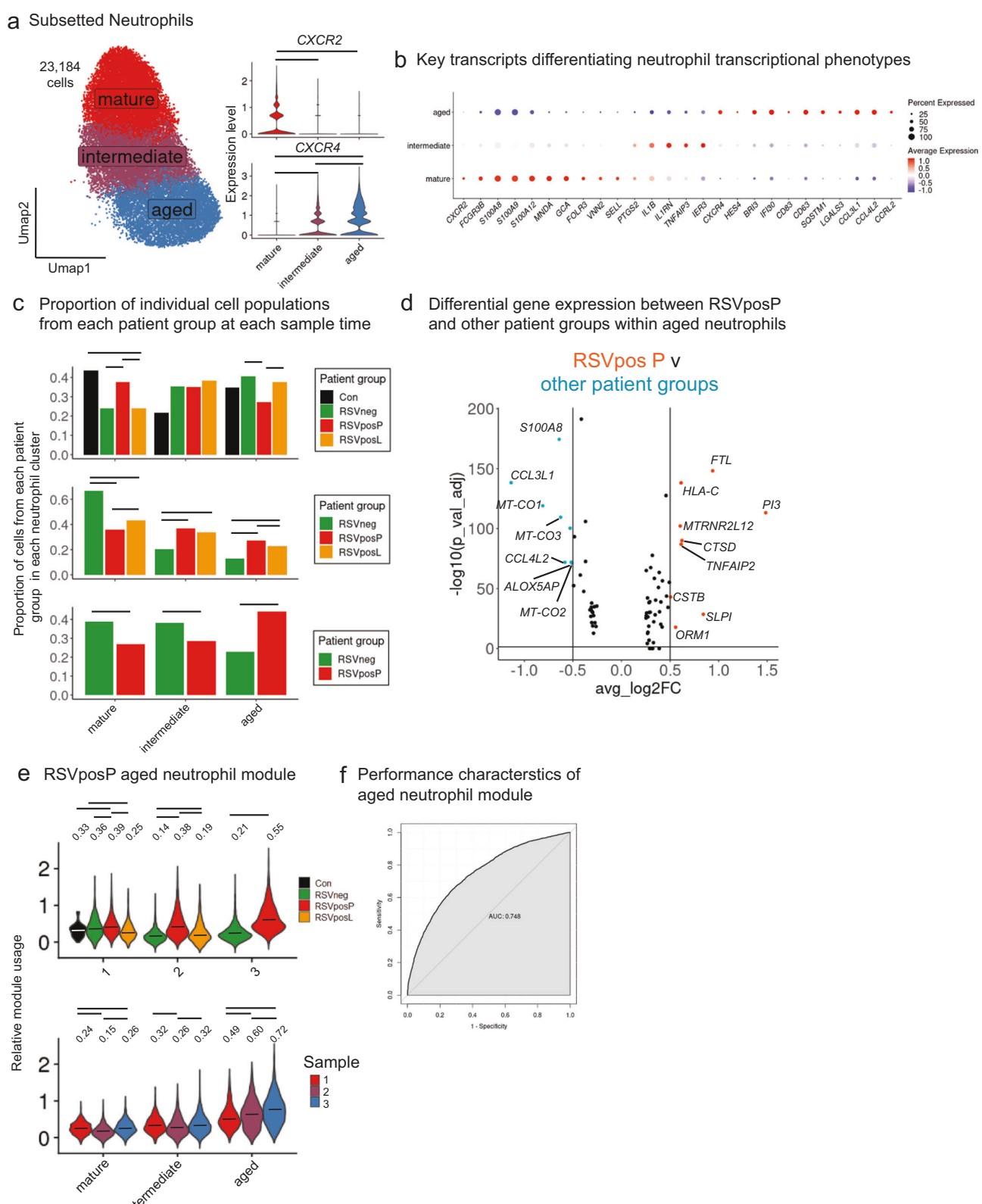

a  Subsetted Neutrophils

b  Key transcripts differentiating neutrophil transcriptional phenotypes

c  Proportion of individual cell populations from each patient group at each sample time

d  Differential gene expression between RSVposP and other patient groups within aged neutrophils

e  RSVposP aged neutrophil module

f  Performance characterstics of aged neutrophil module

patients, characterized by the upregulation of the protease inhibitors *PI3* and *SLPI* as well as *TNFAIP2* (involved in regulation of inflammatory responses) and downregulation of mitochondrial transcripts (*MT-CO1*) and cytokines (*CCL3L1*).

We used the list of top transcripts differentially upregulated by aged neutrophils from RSVposP patient groups for module scoring (Supplementary Data 2). This demonstrated increased module usage

within aged neutrophil clusters in gene expression space that was distinct in distribution compared to previously described neutrophil gene sets (Fig. S4d). There was significantly increased module usage by RSVposP neutrophils at all 3 Sample time points (Fig. 6e, top), and significantly increased module usage within aged neutrophils over time (Fig. 6e, bottom). We finally tested the ability of aged neutrophil module scores to differentiate between RSVposP and RSVposL group

**Fig. 6 | Accumulation of aged neutrophils with a unique transcriptional program in moderate to severe pARDS caused by RSV. a** (Left) UMAP of subsetted neutrophils from the aggregate data set, labeled according to putative age. (Right) Violin plots of *CXCR2* and *CXCR4* expression levels by neutrophil cell clusters. Cell clusters are colored and labeled as in Fig. 6a. Statistical testing for differences between groups was performed using a two-sided Wilcoxon Rank Sum Test with *p*-value adjustment using the Bonferroni correction. Significant differences (below a $p_{adj} < 0.05$) between groups is denoted with a bar (*CXCR2* expression mature_v_intermediate p_adj = 5.9e-107, mature_v_aged p_adj = 0; *CXCR4* expression aged_v_intermediate p_adj = 9.19e-202, aged_v_mature p_adj = 0, intermediate_v_mature p_adj = 2.7e-280). **b** Dot plot of transcripts differentially expressed within neutrophil cell clusters, which are labeled according to Fig. 6a. The size of the dot indicates percent of cells within an individual cluster expressing a particular transcript, and the color indicates the average expression of the transcript within the cell population. **c** Bar charts of proportional abundance of patient group cells within each neutrophil transcriptional cluster (labeled as in Fig. 6a). Testing for significant differences between groups was performed using pairwise proportion test with continuity correction and p-value adjustment using Bonferroni correction (more than two groups compared) and a proportion test (two groups compared). Significant differences (threshold $p_{adj} < 0.05$) between groups are labeled with a bar. Source Data are provided as a Source Data File. **d** Volcano plot demonstrating DEGs between RSVposP and all other patient groups within aged neutrophil populations. DGE testing was performed using Seurat's FindAllMarkers in default settings (two-sided Wilcoxon rank sum test with Bonferroni *p*-

value adjustment). A positive avg_log2FC indicates increased expression in RSVposP aged neutrophils relative to other patient groups. Source Data are provided as a Source Data File. **e** Violin plot of RSVposP aged neutrophil module usage, grouped by Sample time point and split by patient group. Testing for statistical differences included Kruskal-Wallis and pairwise two-sided Wilcoxon rank sum testing followed by *p*-value adjustment using the Benjamini & Hochberg method (for comparisons between more than 2 groups) and an unpaired two-sided *t* test (for comparisons between two groups). Significant differences (below a threshold of $p_{adj} < 0.05$) between groups are indicated with a line above the violin (Sample 1: RSVneg_v_C p_adj = 0.193, RSVposP_v_C p_adj = 0.0094, RSVposL_v_C p_adj = 0.139, RSVposP_v_RSVneg p_adj = 0.013, RSVposL_v_RSVneg p_adj = 7.5e-8, RSVposL_v_RSVposP p_adj<2e-16; Sample 2: RSVneg_v_RSVposP p_adj<2e-16, RSVneg_v_RSVposP p_adj<2e-16, RSVposP_v_RSVposL p_adj<2e-16; Sample 3: RSVneg_v_RSVposP p_adj<2e-16; Mature Sample1_v_Sample2 p_adj<2e-16, Sample1_v_Sample3 p_adj<2e-16, Sample2_v_Sample3 p_adj<2e-16; Intermediate Sample1_v_Sample2 p_adj = 8.6e-14, Sample1_v_Sample3 p_adj = 0.91, Sample2_v_Sample3 p_adj<2e-16; Aged Sample1_v_Sample2 p_adj = 1.1e-10, Sample1_v_Sample3 p_adj<2.2e-16, Sample2_v_Sample3 p_adj<2e-16) and median module usage is indicated with a bar in the violin and further annotated numerically above the violin. Source Data are provided as a Source Data File. **f** Receiver Operator Characteristic (ROC) curves for RSVposP aged neutrophil gene module differentiating between RSVposP and RSVposL group neutrophils. AUC denotes Area Under the ROC curve.

---

neutrophils by generating a ROC curve (Fig. 6f). We found an AUROC score of 0.75, which further supports specificity of this gene set for RSVposP neutrophils.

### FOLR3 as a marker of disease severity in pARDS

To identify early transcriptional markers of pARDS, irrespective of symptomatology and TA cell type, we grouped patients according to the primary study cohort into which they were enrolled at the Sample 1 time point (Fig. 7a). We first performed differential gene expression testing between P and L cohorts in Sample 1 cells (Fig. 7b). This analysis identified differential upregulation of *VCAN*, mitochondrial transcripts (*MT-CO2*), ribosomal genes (*RPL34*), and *FOLR3*, and differential downregulation of cytokines (*CCL3L1*, *CCL3*, *IL6*) and ISGs (*ISG15* and *IFITM3*) in P-cohort TAs.

We then analyzed expression of these transcripts between the three study groups in Sample 1 TAs and confirmed upregulation of *FOLR3*, *S100A12*, and *VCAN* in P-cohort TAs and *ISG15*, *CCL4*, and *CCL3* in L-cohort TAs. While expression of mitochondrial and ribosomal transcripts was higher in P-cohort relative to L-cohort TAs, expression of these were higher in C-cohort TAs (Fig. 7c). Further analysis demonstrated specificity of *FOLR3* expression by neutrophils, *S100A12* by neutrophils and monocytes, *VCAN* by monocytes and "inflammatory" macrophage populations, *ISG15* by Ma_ISG cells, and cytokines (*CCL3L1*, *CCL3*, *CCL4*) by neutrophils and mononuclear phagocytes (Fig. 7d).

We then measured concentrations of proteins encoded by these transcripts in TA supernatants obtained from an independent cohort of patients enrolled at another children's hospital (Fig. 7e and S5). Samples included in this analysis were chosen to match our age and pre-admission health status inclusion criteria. We assigned disease severity to patients in this cohort using the same consensus definitions[16] (Supplementary Data 3). We found that concentrations of FOLR3 were significantly higher ($p = 0.04$) in first sample TAs obtained from patients with moderate to severe pARDS compared to patients with no or mild pARDS. Additionally, we identified a non-significant trend towards increased concentrations of Resistin ($p = 0.054$) and S100A12 ($p = 0.15$) as well as decreased concentrations of Interferon alpha ($p = 0.15$) in patients with moderate to severe pARDS compared to patients with no or mild pARDS.

Together, these analyses suggest severity-associated responses detectable in TA samples from patients with pARDS that are consistent

with a neutrophil predominant response in patients with moderate to severe pARDS and a MNP predominant response in patients with no or mild pARDS.

## Discussion

This study details single cell transcriptional profiles in TA samples obtained from infants with ARF requiring IMV. Consistent with the known epidemiology of ARF in this age group[11], we identified viral LRTI, and in particular RSV-associated LRTI, as the most common cause of ARF in our cohort. Our analyses suggest etiology-dependent and illness severity-dependent alterations of TA cellular composition and transcription in epithelial cells, MNPs, and neutrophils. We present several findings that provide insight into the pathogenesis of ARF in infants.

We characterized RSV expression using two distinct methods, including by viral capture sequencing and by RSV expression in single cell transcriptomic data. By ViroCap sequencing, there was a trend towards more TA supernatant reads mapping to RSV in patients with no or mild pARDS related to RSV compared to those with moderate to severe pARDS related to RSV. In our single cell transcriptomic data, we consistently identified a significantly higher proportion of RSVposL cells infected with RSV (including epithelial cells, MNPs, and neutrophils) compared to RSVposP cells; however, RSV expression, represented as either a normalized abundance or as a proportion of the cellular transcriptome, was not statistically different between groups, but was generally higher in RSVposP cells. RSV viral load (VL) has been associated previously with disease severity, albeit with conflicting results depending on the patient population and study design[35–37]. Our study suggests that host response, as opposed solely RSV VL, makes as an important contribution to differences in illness severity.

In keeping with this interpretation, divergent severity-associated epithelial responses to ARF are apparent within our data. Amongst subclustered epithelial cells, we identified significantly increased ISG gene module[29] usage by as well as interferon and inflammatory response pathways within RSVposL compared to RSVposP epithelial cell populations. Gene lists generated using DGE testing to identify differences between P and L cohort epithelial cells had AUROC scores of 0.70 to 0.78, respectively, suggesting these gene modules are potentially informative classifiers. These gene lists include transcripts involved in epithelial defense (SCGB3A1) and cellular detoxification (*GPX4*, *GSTP1*, *PRDX5*, and *SERF2*) for the P cohort gene list and

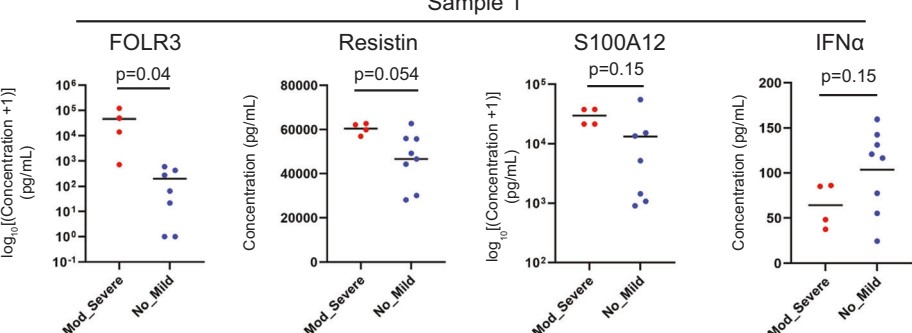

**Fig. 7 | Identification of FOLR3 as a potential airway marker of severity in pARDS. a** UMAP plots of Sample 1 cells in aggregate split and colored by Patient cohort of origin. **b** Volcano plot demonstrating DEGs between P and L Patient cohort Sample 1 TA cells. DGE testing was performed using Seurat's FindAllMarkers in default settings (Wilcoxon rank sum test with Bonferroni p-value adjustment) comparing Sample 1 P-cohort cells and Sample 1 L-cohort cells (a positive avg_log2FC indicates increased expression P-cohort cells and a negative avg_log2FC indicates increased expression in L-cohort cells). Source Data are provided as a Source Data File. **c** Dot plot of demonstrating expression of transcripts identified in Fig. 7b by each Patient cohort at the Sample 1 time point. The size of the dot indicates percent of cells from an individual patient cohort expressing a particular transcript, and the color indicates the average expression of the transcript within

the cell population. **d** Dot plot demonstrating expression of transcripts identified in Fig. 7b by each cell population at the Sample 1 time point. The size of the dot indicates percent of cells from an individual cell population expressing a particular transcript, and the color indicates the average expression of the transcript within the cell population. **e** Concentrations of select analytes measured in tracheal aspirate samples from patients enrolled in another study at a different institution ($n = 4$ patients for Mod_Severe, $n = 7$ patients for No_Mild analytes measured by ELISA and $n = 8$ for analytes measured by Luminex). Statistical testing was performed between groups using an unpaired two-tailed $t$-test. P-values for statistical comparisons between groups are noted in the plots. Source Data are provided as a Source Data File.

monocyte chemoattractants (*CCL3/4*), ISGs (*IFI6*, *ISG15*, and *IFITM3*), and *TIMP1* (an inhibitor of metalloproteinases) for the L cohort gene list. Together, these data suggest epithelial cells, and potentially interacting innate immune cells, from lower illness severity TAs (RSVposL) are coordinating a type I interferon response to infection, whereas epithelial cells from patients with higher illness severity are less interferon-responsive and are responding to mitochondrial and oxidative stress. These findings agree with previous nasal transcriptome studies of RSV disease severity, in which patients with more severe disease demonstrated lower ISG expression[12,13] and lower viral loads compared to a comparator group demonstrating a less severe disease phenotype[13].

We identified illness severity and etiology-dependent differences in MNP transcriptional responses to ARF. We were most intrigued by the significantly higher proportional abundance of macrophages expressing repair-associated transcripts (i.e., *AREG*), the scavenger receptor involved in recognition of apoptotic cells *OLR1*[38], *IL10*, and the glycoprotein *THBS1* in Sample 2 TAs obtained from patients in the RSVposL group compared to patients in the RSVposP and RSVneg groups. Overall, this gene expression profile suggests a regulatory role played by this macrophage population, potentially with a particular focus on phagocytosis of apoptotic cells. IL-10 is a key anti-inflammatory cytokine that reduces excessive immunopathology, potentially at the cost of incomplete pathogen control in certain contexts[39]. IL-10 has been implicated in human ARDS severity[40,41] and has been shown to regulate neutrophil chemotaxis to the lung in a preclinical model of lower respiratory tract bacterial infection[42]. Macrophages upregulate IL-10 following exposure to proinflammatory mediators (i.e., lipopolysaccharide (LPS)) as well as after contact with apoptotic cells[43]. IL-10 producing macrophages have increased phagocytic capacity with preferential specificity for early apoptotic cells[38,44]. Thrombospondin-1, an extracellular matrix glycoprotein recognized by CD36 participates in interactions between phagocytes and apoptotic cells. Indeed, in a preclinical ARDS model, *Thbs1*[−/−] mice demonstrate impaired resolution of pulmonary inflammation characterized by reduced macrophage IL-10 production, increased inflammatory cytokine expression, and exacerbated neutrophilic pulmonary infiltrates[45]. The oxidized low-density lipoprotein receptor 1 (LOX1; encoded by *OLR1*) is another scavenger receptor expressed by phagocytes (in addition to CD36) involved in recognition of apoptotic cells[38]. Intriguingly, *OLR1* expression is significantly upregulated in dendritic cells generated in vitro from peripheral blood monocytes in the presence of interferon alpha[46]. As noted above, we consistently identified increased ISG module usage across multiple cell types in RSVposL TA samples. Taken together, we hypothesize that this regulatory macrophage population, which is enriched in TAs from patients with lower illness severity potentially due to increased type I interferon signaling, plays a key role in mediating resolution of inflammation via clearance of early apoptotic cells and reduction of neutrophil chemotaxis to the airways.

Airway neutrophilia has previously been demonstrated in the context of critical RSV and is thought to play a role in disease severity[14,47]. Consistent with this, we observed a progressive increase in proportional representation of neutrophils across sampling times within pARDS patients. Aged neutrophils were enriched in RSVposP patient TAs and demonstrate a unique transcriptional profile, enriched with protease inhibitors (i.e., *SLPI*, *PI3*, and *CSTB*), *ORM1* (an acute phase reactant of uncertain immunoregulatory function), and *TNFAIP2* (an inhibitor of TNF signaling). Aged neutrophils, which upregulate CXCR4, have been shown previously to demonstrate inflammatory responses via Toll-like Receptor 4 (TLR4) and Integrin signaling compared to non-aged neutrophil phenotypes[48]. Using an RSVposP aged neutrophil gene list, we identified a AUROC of 0.75, which suggests that this module may perform well as a classifier of disease severity amongst infants with RSV infection. Together, these data support the

conclusion that moderate to severe RSV-associated pARDS is associated with the accumulation of aged neutrophils displaying a unique transcriptional phenotype, which may contribute to the development of clinically severe disease.

We identified increased expression of *FOLR3* in patients with pARDS compared to patients with no or mild pARDS and provide initial potential validation of tracheal aspirate FOLR3 protein concentration as a marker of pARDS severity using samples obtained from an independent study at another hospital. *FOLR3* encodes the secreted Folate Receptor γ, one of three human folate receptors. It is known to be a component of neutrophil secondary granules[49] and is the predominant folate receptor detectable in plasma[50]. As it has no murine orthologue, less is known about FOLR3 function in the context of an immune response; however, it may contribute to antimicrobial defense via binding and depletion of extracellular folate[50]. Along with validation in a larger cohort, future study of whether FOLR3 concentrations are a marker of increased TA neutrophil numbers or an indicator of increased secondary granule production and degranulation may be of interest for future study.

Due to the dynamic nature of clinical treatment and illness presentation of ARF, we note unavoidable limitations to our study. TA samples, which contain mucous secretions, RNAses, and extracellular nucleic acid, inherently present technical difficulties with respect to single cell RNA sequencing. Thus, obtaining single cell suspensions free of mucous and extracellular nucleic acid is potentially problematic. To mitigate these potential technical issues, we processed all samples immediately and utilized a Percoll gradient to purify cells from mucus. Despite these efforts, we acknowledge that these unavoidable technical limitations may have contributed to less-than-ideal QC metrics for several samples included in the final dataset (Supplementary Data 4). A potential explanation for observed differences in gene and RSV expression between groups hinges upon differences in time from symptom onset to enrollment, a known limitation for studies of naturally acquired infections in humans. However, the relevant findings are generally consistent with our current understanding of ARDS-associated immunopathology. Finally, there were differences in demographic variables, including age, sex, and race, between patient cohorts, which may contribute to differences in immune responses between individuals. Thus, our findings, which were developed using analyses of samples obtained from patients admitted to a single center, will require future validation in a larger cohort in which these demographic variables can be better balanced between comparator groups.

Despite these limitations, this study details unique single cell transcriptomic characterization of responses to ARF occurring in the infant airway. A particular strength of this study is the prospective design, wherein patients contributed samples longitudinally over the course of the first week of hospitalization. We identified illness severity, etiology, and sample time point dependent changes in cell composition and transcriptional response, even in patients presenting similarly clinically. Our most salient findings are reduced interferon stimulation within epithelial and MNP cell populations, reduced proportional abundance of *IL10* expressing regulatory macrophages, and increased airway neutrophilia associated with a unique transcriptional profile in patients with moderate to severe pARDS compared to patients with no to mild pARDS. We go on to identify increased FOLR3 concentrations in TAs collected in an independent study at another center. Together, these data implicate type I interferons, macrophage-mediated clearance of early apoptotic cells, and neutrophil ageing as important contributors to disease severity in infants with RSV-associated ARF requiring endotracheal intubation and IMV.

## Methods
### Patient recruitment and ethics approval
Patients were enrolled in this prospective observational cohort study conducted at LeBonheur Children's Hospital in Memphis, Tennessee

from March 2018 to February 2020. This study was approved by the University of Tennessee Institutional Review Board (IRB) (IRB Number 17-05744-XP STJUDE) and the St. Jude Children's Research Hospital IRB (IRB Number Pro00008351). Signed informed consent for participation was obtained from the patient's parent or legal guardian prior to inclusion in the study.

### Inclusion and exclusion criteria

Children aged 0–12 years with acute respiratory failure requiring endotracheal intubation and invasive mechanical ventilation (IMV) were enrolled in one of three groups: patients with a known or suspected lower respiratory tract infection (LRTI) who met consensus criteria for the pediatric acute respiratory distress syndrome (pARDS)[16] (P cohort), patients with a known or suspected LRTI who do not meet consensus criteria for pARDS[16] (L cohort), and control patients with respiratory failure without known or suspected LRTI or findings consistent with a diagnosis pARDS[16] (C cohort). Consensus criteria were applied based on review of the clinical history, results of clinically obtained laboratory tests, and results of chest imaging by the principal investigator (TF) in consultation with the pediatric intensive care unit (PICU) team managing the patient's care. Patients were excluded from enrollment if they had previously been diagnosed with chronic lung disease of prematurity, had not been discharged from the hospital following delivery, uncorrected congenital heart disease, pulmonary hemorrhage, and primary or secondary immunodeficiency. Patients were additionally excluded if they were prescribed inhaled or systemic corticosteroids prior to hospitalization in the previous four weeks, had a tracheostomy, required home supplemental oxygen, or required home non-invasive mechanical ventilatory support. Finally, patients were excluded if they required extracorporeal membrane oxygenation (ECMO) support prior to being approached for enrollment.

### Study design

We designed the study to collect TA samples from enrolled patients at one of three timepoints. The timing of sample collection depended on the cohort into which they were enrolled based on the a priori assumption that P-cohort patients would have the longest duration of IMV (Fig. 1a). Three TA samples were collected from patients enrolled in the P-cohort. The first was between 6 and 24 h after endotracheal intubation (Sample 1), the second was 48–72 h after Sample 1 (Sample 2), and the third was 48–72 h after Sample 2 (Sample 3). Two TA samples were collected from patients in the L-cohort at the same time as Samples 1 and 2 in the P-cohort. One TA sample was collected from patients in the C-cohort at the same time as Sample 1 in the P and L-cohort. At the same time as sample collection, patient data, including demographics, ventilator settings, results of microbiologic testing, and interventions, including antibiotics, corticosteroids, and inhaled nitric oxide, was abstracted from the electronic medical record (EMR) and stored using Microsoft Excel. Patients were followed for 60 days after enrollment to determine duration of IMV, PICU length of stay (LOS), and PICU mortality. Patient sex was not considered in the study design; however, these data were collected as listed in the EMR for the patient after obtaining informed consent for participation on the study.

### Sample collection

After enrollment, tracheal aspirates (TAs) were collected by the patient's respiratory therapist or registered nurse as per routine practice and timing for routine endotracheal tube care. This was intended to obtain samples that would be comparable to those able to be obtained in the course of standard patient care. 2–4 milliliters (mL) of sterile saline were instilled into the patient's endotracheal tube (ETT). The TA was then collected using in-line endotracheal tube suctioning into a sterile specimen suction trap at the time when the patient's ETT would have otherwise needed to be cleared for routine

care. After collection, the specimen was placed on ice and immediately transported to be processed fresh for downstream assays (Fig. 1b).

### Sample processing

The TA was first transferred from the sterile specimen suction trap to a 15 mL conical tube. After pipetting the sample into the 15 mL conical tube, 1 mL of phosphate buffered saline (PBS) was used to rinse the sterile specimen suction trap and then added to the 15 mL conical tube. Samples were then centrifuged at 500 × g for 5 min at 4 degrees Celsius (°C). The supernatant was collected into a 15 mL tube and mixed thoroughly before dividing into 500 μL aliquots and stored at −80 °C for later nucleic acid extraction for microbiologic sequencing (see below). We initially found that mucous in the samples complicated achieving a clean single cell suspension for single cell RNA-seq protocols. To remove mucous and other debris, we used a Percoll gradient consisting of four different concentrations. First, 100% Percoll was added to a 15 mL conical tube. A 60% Percoll gradient was overlaid on top of the 100% Percoll. After the initial centrifugation step and removal of supernatant, the pellet was resuspended in 40% Percoll and overlaid on top of the 60% layer. Finally, a 30% Percoll layer was overlaid on top of the 40% Percoll layer. The Percoll gradient (60/40/30) containing the resuspended pellet in the 40% layer was centrifuged at 2000 × g for 20 min at room temperature without a brake. After centrifugation, cells were collected from the 60/40 and the 40/30 interface and transferred to a new 15 mL conical tube. Percoll was diluted with PBS to a 15 mL total volume and centrifuged at 500 × g for 5 min at 4 °C. The supernatant was removed and the pellet was resuspended in 1–2 mL of PBS and 0.04% ultrapure bovine serum albumin (BSA). This suspension was then filtered through a 30 μm cell strainer to remove any residual debris. At this point, cells were counted using a hemocytometer with filtered 0.4% trypan blue serving as a viability count. After confirming successful creation of gel bead in emulsion (GEM) in the initial step of single cell gene expression library preparation (described below), any cells remaining were frozen down at 1.5–5 million cells/mL in BAMBANKER Freezing Medium (Fujifilm Wako Chemicals Corporation, Richmond, VA).

### Single-cell gene expression library preparation

Libraries for single cell gene expression analyses were generated individually on the fresh TA single cell suspensions generated as above using the 3′ Gene Expression Kit (version 2, PN 120235 (Gel Bead Kit) and PN 120264 (Library Kit), 10X Genomics, Pleasanton, CA) according to the manufacturer's instructions. Because cell yield varied by sample, we targeted final single cell suspensions to a concentration of 1000–1200 cells/μL and then loaded up to 17,000 cells onto a 10X Genomics chip for each sample. Due to a technical failure during loading of the freshly collected sample, Sample 1 from patient P1 was rerun using cryopreserved cells. This was the only sample processed and analyzed using cryopreserved cells. Libraries were sequenced on the Illumina NovaSeq platform (Illumina, San Diego, CA).

### Single-cell analysis

Sequencing data were processed using CellRanger (v6.0.0, 10X Genomics) with a custom reference that included the human genome (GRCh38-2020-A; 10X Genomics) and the genomes of Enterovirus B (NC_038307.1), Enterovirus D (NC_038308.1), Respiratory syncytial virus (NC_038235.1), Rhinovirus A (NC_038311.1), Rhinovirus B (NC_038312.1), Rhinovirus C (NC_038878.1), Torque teno mini virus (NC_020498.1), Torque teno virus (NC_015783.1), Adenovirus B (NC_011203.1), Adenovirus C (NC_001405.1), and each segment of Influenza A (A/Michigan/45/2015(H1N1)). Quality control metrics, including median number of genes per cell and fraction of reads within cells, were recorded and are reported in Supplementary Data 4. Due to the heterogeneous nature of samples, the independent libraries were subsequently aggregated using CellRanger, without normalization,

into a single matrix. Downstream analyses were performed using the Seurat (v4.0.3) framework[51].

To exclude potential doublets, we removed cells that exhibited UMI counts or feature counts greater than the 99.5% probability quantile for each sample. Cells with greater than 20% of gene expression owing to mitochondrial genes were excluded as putatively dead or dying cells. We inferred cell cycle using the Seurat CellCycleScoring algorithm and as described elsewhere[52]. Each library was normalized using the SCTransform algorithm, with percent of mitochondrial expression and cell cycle effects regressed out. The fastMNN algorithm[53] was then utilized to integrate datasets from distinct libraries, effectively minimizing patient- and sample-specific differences in order to identify similar transcriptional subsets; this approach was repeated independently across all major cell subsets, including epithelial cells, MNPs, and MNPs with neutrophils. In each case, the first 15 fastMNN dimensions were used for UMAP dimensionality reduction and for nearest-neighbor graph construction for identifying transcriptional clusters in Seurat.

Differential gene expression (DGE) testing was performed between patient groups using the FindAllMarkers function in the *Seurat* R package with default settings, which identifies differentially expressed genes between two groups of cells using a Wilcoxon Rank Sum test. Significant differences between cell groups were determined using adjusted p-values, which are adjusted using a Bonferroni correction using all genes in the dataset[51]. Gene lists were generated from previously published manuscripts[29,30,32,33] and were used to generate module scores using Seurat's AddModuleScore. The only change that was made to any gene list was the addition of *IFI6* to the ISG module described by Wilk et al. We performed this analysis with the original gene set and with the addition of *IFI6* and reached the same conclusion with regard to differences in module usage between etiology and illness severity patient groups. Pathway enrichment analysis was performed with Seurat's DEenrichRplot command, which returns bar plots with top enriched terms from the selected EnrichR database. For these analyses, the enrich.database was set to "MSigDB_Hallmark_2020", max.genes was set to 2000, and num.pathway was set to 10. ROC curves and AUROC scores were generated from module scores using the R package pROC (v1.18.0). We implemented CellChat (v1.1.3)[24] by first extracting CellChat input files from our Seurat object and subsequently creating a CellChat object. We set the ligand-receptor interaction database to CellChatDB.human for these analyses with no changes made to the database. This is a database of ligand-receptor interactions taken from previously published literature or from the Kyoto Encyclopedia of Genes and Genomes (KEGG) and includes 1939 validated molecular interactions. We performed all analysis using the standard workflow. To identify differences in ligand-receptor interactions between groups, we created a CellChat object for each patient group and merged them using the mergeCellChat command. Subsequent comparisons between groups followed the standard workflow.

## ViroCap enrichment and sequencing

RNA and DNA were extracted from cryopreserved TA supernatants using the quick-DNA/RNA Miniprep Plus Kit (Catalog Number D7003; Zymo Research, Irvine, CA) according to the manufacturer's protocol in two batches. Briefly, between 200 μL and 450 μL of TA supernatant (depending on initial yield after sample processing; 44 of 51 (86%) TAs started with 450 μL of TA supernatant) was combined with 300 μL to 550 μL (to a uniform starting total volume of 750 μL) of RNA/DNA Lysis Buffer and centrifuged using the Spin-Away Filter in a Collection Tube. 450 μL of nuclease free water combined with 300 μL of RNA/DNA Lysis Buffer was used as a negative control for each batch of extractions. The Spin-Away Filter was placed in a new collection tube and stored at 4 °C while performing RNA purification. For this, 200

proof ethanol (ETOH) (Sigma Aldrich, St. Louis, MO) was added to the flow-through from the Spin-AwayFilter centrifugation step. 750 μL of the mixture of ETOH and flow-through was added to the Zymo-Spin IIICG Column in a collection tube and centrifuged. The flow-through was discarded and the remaining 750 μL was added to the same Zymo-Spin IIICG Column and centrifuged again. DNA and RNA were then prepared and washed according to the manufacturer's protocol and eluted into a final volume of 30 μL of water. DNA and RNA were quantified using Qubit High Sensitivity dsDNA and RNA Assay Kits (Catalog Number Q32851) on a Qubit 4 Fluorometer (ThermoFisher, Waltham, MA). Between two and four nucleic acid extractions were performed for five of the TA samples (P003_0, P003_2, P005_1, P005_2, P006_2) due to low initial RNA yield.

Double stranded cDNA (dscDNA) was created from total RNA extracted and purified from TA supernatants using the Maxima H Minus Double-Stranded cDNA Synthesis Kit (Catalog Number K2562; ThermoFisher Scientific, Waltham, MA) according to manufacturer's protocol. Briefly, 13 μL of total RNA was added to 1 μL of random hexamer primer for first strand cDNA synthesis. Following second strand cDNA synthesis, 100U of RNase I was added prior to proceeding with dscDNA cleanup. A 1.8X AMPure (Beckman Coulter, Brea, CA) bead cleanup was used for dscDNA purification and the purified product was eluted into a final volume of 40 μL in Buffer EB (Qiagen, Hilden, Germany). Double-stranded cDNA concentrations were then quantified using Qubit High Sensitivity dsDNA Assay Kits (Catalog Number Q32851) on a Qubit 4 Fluorometer (ThermoFisher, Waltham, MA). Double-stranded cDNA was created and purified from total RNA from each aliquot of TA supernatant for these samples. Following dscDNA purification, dscDNA from each sample was combined and concentrated using a 1.8X AMPure XP (Beckman Coulter, Brea, CA) cleanup into 10 μL Buffer EB (Qiagen, Hilden, Germany).

Double-stranded cDNA and genomic DNA were each used to construct sample-specific libraries with KAPA HyperPrep kits (Catalog Number 07 962 363 001; Roche) with mechanical shearing. Approximately 100 ng of each individually indexed library was pooled by nucleic acid type (cDNA or gDNA), with up to 14 samples in a pool, and pools were enriched for viral content using the SeqCap EZ Developer kit (Roche) designed with ViroCap probes (IRN40000[20];). Library generated from purified Seal Influenza A hemagglutinin cDNA was used as a positive control. Pools were blocked with human COT and IDT blockers before a 20-h hybridization. Enriched library pools were sequenced on the Illumina NovaSeq platform (Illumina, San Diego, CA) at 100 × 100 base pairs.

## ViroCap analysis

Raw reads were concatenated by sample and trimmed using Trimmomatic[54] to remove sequencing adapters. Kraken2[55] (v2.0.7b) was used to taxonomically classify reads using the Standard reference (which includes archaea, bacteria, viral, plasmid, human, and Uni-Vec_Core sequences) downloaded from the Kraken website in September of 2020. Bracken[56] (v2.5) was subsequently used to estimate relative abundances of taxonomic annotations identified by Kraken2. Additionally, Kraken2 annotations were processed with Pavian (v0.8.4)[57] to summarize the abundance of classified reads.

## Endotracheal Aspirate Protein Quantification

All samples were stored at −80 °C before they were diluted 1:2 for Cytokine analysis using Human Magnetic Luminex Assay (PN LXSAHM-19; R&D Systems) with the manufacturer's set protocol. Data was collected on a Luminex 200 (Luminex) and analyzed on Milliplex analyst software (Vigene tech). In addition, after 1X thaw, samples were tested with individual Quantitative ELISA Kit per manufacturer's protocol as follows: Human Thrombospondin 1:20(PN ab193716; Abcam); SLP-I

1:20(PN DP100; R&D Systems); Annexin A1 1:20(PN ab222868; Abcam); IL-8/CXCL8 1:20(PN D8000C; R&D Systems); CCL20/MIP-3 alpha 1:20(PN DM3A00; R&D Systems); CXCL10/IP-10 1:50(PN DIP100; R&D Systems); Human EN-RAGE 1:100(PN D41052-05; R&D Systems); Human S100A8/S100A9 Heterodimer 1:1000(PN D48226-05; R&D Systems). After 2× thaw samples were diluted for another set of individual Quantitative ELISA Kits as follows; Human FOLR3 1:20(PN DY5319; R&D Systems); Human CCL3L1/LD78 Beta 1:20(PN NBP2-7065; Novus); Human CCL4L1/LAG-1 1:20(PN NBP2-75067; Novus). All ELISAs were read on ELISA Reader (Molecular Probes) using dual wavelengths of 450/540 and data was analyzed using SOFTMAX Pro software 7.1.4.

### Analysis of endotracheal aspirate protein concentrations

For measurements above the upper limit of detection (ULOD) or below the lower limit of detection (LLOD) in the Luminex multiplex assay we used the value of the ULOD or the LLOD in our analysis. For ELISA-based measurements we averaged the analyte concentrations extrapolated from the standard curve for Samples, which had been run in duplicate. Samples for which there was no fluorescence detected in either well were treated as not detected (value of 0 pg/mL). We noted that even at a 1:20 dilution the concentrations of SLPI exceeded the range of the standard curve. Concentrations were plotted and statistical testing between patient groups was performed using Graph Pad Prism (version 8.4.3) Software.

### Reporting summary

Further information on research design is available in the Nature Portfolio Reporting Summary linked to this article.

## Data availability

The raw sequencing data generated in this study have been deposited in the Sequence Read Archive database under accession code PRJNA971535. All other data are available in the article and its Supplementary files or from the corresponding author upon request. Source data are provided with this paper.

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

## Acknowledgements

This study was funded by a Children's Infection Defense Center Grant (St. Jude Children's Research Hospital) awarded to Paul Thomas and Tim Flerlage, LeBonheur Children's Hospital Clinical Fellow Research Award to Tim Flerlage, AI121832 and AI144616 to Paul Thomas, ALSAC at St. Jude to Paul Thomas and Tim Flerlage, AI154470 to Paul Thomas and Adrienne Randolph, and PICFlu Study NIH AI084011 to Dr. Adrienne Randolph. The funders had no role in the design or analysis of the study.

## Author contributions

T.F. wrote and designed the study, enrolled the patients, processed samples, performed single cell assays, analyzed the data and prepared the manuscript. J.C.C performed data analysis and contributed to manuscript and figure preparation. E.K.A. contributed to single cell and ViroCap assays and manuscript preparation. D.S. contributed to data collection and manuscript preparation. S.T. contributed to ViroCap data analysis. S.S. performed protein concentration measurements in tracheal aspirate samples provided by collaborators at Boston Children's Hospital (T.N. and A.R.). G.R. contributed to ViroCap assays and manuscript preparation. T.N. provided samples and associated clinical data for external validation cohort from Boston Children's Hospital and contributed to manuscript preparation. A.R. provided samples and clinical metadata for external validation cohort from Boston Children's Hospital and contributed to manuscript preparation. A.N.W. contributed to study design, patient identification and enrollment, and manuscript preparation. P.G.T. oversaw the study, contributed to data analysis, manuscript preparation, and figure preparation.

## Competing interests

The authors declare no competing interests.
