## [Peer Review File · Nature Communications]

Single Cell Transcriptomics Identifies Distinct Profiles in Pediatric Acute Respiratory Distress SyndromeEditorial Notes:

This manuscript has been previously reviewed at another journal that is not operating a transparent peer review scheme. This document only contains reviewer comments and rebuttal letters for versions considered at *Nature Communications*.

REVIEWER COMMENTS

Reviewer #1 (Remarks to the Author):

We appreciated the chance to review the revisions to Flerlage et al. In large part, we feel that the authors were very responsive to prior suggestions outline in the initial review. In particular, we felt the figures were much clearer and easier to interpret than the original version and the inclusion of experimental limitations in the conclusions was helpful. We likewise appreciated the new analyses included in the manuscript. The quality of the dataset and accompanying analyses is high, and will undoubtedly provide an invaluable resource for specialists in a wide variety of fields. While enthusiasm for eventual publication is high, there remain some concerns about the presentation of the data.

Major point:

The key conclusions that the authors wish the reader to take away are obfuscated by the vast array of complementary findings that are simultaneously reported. As stated from their conclusions, "Our most salient findings are reduced interferon stimulation across multiple TA cell types, reduced proportional abundance of IL-10 expressing regulatory macrophages, and increased airway neutrophilia associated with a unique transcriptional profile in subjects with moderate to severe pARDS compared to subjects with no to mild pARDS." While the richness of the dataset is fascinating, many of the reported results are not directly related to these key findings and ultimately distract from their appreciation. For example, both the virocap data and description of the FOLR3 finding (while interesting) are not essential for their key conclusions. These interesting auxiliary findings that are not central to their key results would be better suited for a separate manuscript or a supplemental results section.

Minor point:

--Please median line to violin plots. For some data, it is not clear which groups are higher than others.
--More detailed titles and labeling of figure panels would be helpful (eg Fig S5).

Reviewer #2 (Remarks to the Author):

Despite extensive revisions and redesigned analyses the manuscript is not substantially improved. It remains descriptive due to the small sample size, heterogeneity of patient groups by age, duration of illness, and lack of clarity regarding the infecting pathogens conclusions are unclear. Antecedent risk factors and treatments have not been carefully documented. The range of statistical/bioinformatics analysis is complex and poorly validated. Conclusions are not supported by orthogonal experimentation and are predominantly speculative. Because of the study design, lack of appropriate controls, and complexity of bioinformatic analyses, the conclusions drawn from the complex data set are limited.

Specific concerns:

1) This is a diverse, heterogenous group of patients. Thus, the small study population remains a

significant concern, as other reviewers expressed. This problem has not been addressed. The severe cohort was an average age of 5 months; the milder cohort was an average age of 1 month; the controls were ten months of age. The first year of life is marked by dynamic changes in immune maturation (see Olin et al., PMID: 30142345). An appropriate selection of age-matched cohorts is needed to support meaningful comparisons among the groups.

2) A larger and as yet unresolved concern is that data linking severity with transcriptomic features are not supported by robust statistical analysis or mechanistic studies that validate the significant conclusions.

3) The new inclusion of analysis with CellChat and SCENIC provides little knowledge related to ARDS. For example, communication pathways associated with thromboinflammation (thrombospondin and c-type lectin) were enriched in RSV pol L pathways (less severe). This seems counterintuitive as one would expect significant thrombosis and microvascular damage in clinically severe ARDS (reviewed in PMID: 27997925). Similarly, activation of IL1, cell apoptosis, and CCL signaling (pathways associated with antiviral responses) were more robust in L (less severe) group compared to the P (more severe groups), again seemingly counterintuitive.

4) The authors have identified > 12 epithelial clusters, some of which represent canonical cell types. However, several clusters, for example, Epi_Int_RSV_1..... Epi_MT_High and Immune_HES likely represent transitory transcriptional states rather than distinct cell types and perhaps an artifact of over-clustering. It may be more informative to perform a pseudobulk RNA analysis with more robust DESEQ2-based statistics (which the authors can do, given the multiple samples in each experimental group). Regardless of the analysis, the inference that epithelial responses to LRTI-associated pARDS depend on severity remains untested. This could be easily accomplished by identifying a 'unique transcriptomic signature in the epithelium in P and L cohorts and then calculating a ROC based on 'the defined transcriptomic signature' and disease severity as done by Wilk et al.

5) The authors suggest that epithelial cell clusters expressing RSV genes are innate immune cells, which may have been co-captured in oil emulsion. While that certainly is possible, it is more likely that these represent contamination from more than one cell. How well were doublets accounted for in QI analysis; such a lineage switch has not been carefully validated is unprecedented in the literature.

6) Engagement of different inflammatory programs (Interferon/apoptosis vs. cytokine/immune cell accrual) by RSV- infected cells and bystander cells is expected and not novel. Notwithstanding the limited cell yields and the challenges of studying vulnerable populations, there is a need for orthogonal validation of major conclusions (FACS or exvivo analysis). Furthermore, no experimental data support the concept that bystander cells coordinate immune responses and that bystander cells relieve intracellular stress.

7) Could the increased proportion of RSV-infected epithelial cells in the less severe groups be due to cell loss (dying cells not captured or discarded during QI)?

8) Developmental trajectories directed from epithelial cell populations through the virus hi cluster to MNP clusters (Fig. 4i) is confusing and unprecedented. Do the authors imply that epithelial cells differentiate into MNP, or are the source of the epithelial cells as an 'inflammatory signaling wave' that propagates to MNP?

9) Identifying distinct MNP types with pro-repair and inflammatory phenotypes could explain the differences in clinical severity. Unfortunately, the authors have not robustly linked transcriptional reprogramming of MNP with disease severity (an example as done by Wilk).

10) Accumulation of aged-hyperinflammatory neutrophils in P (severe) vs. L (less severe) groups, while novel in the context of pediatric LRTI, has been shown and validated orthogonally by FACS by several groups in human and animal models.

Reviewer #3 (Remarks to the Author):

I do think the manuscript is improved vs the first round, but still suffers from being quite long and a little diffuse. The paragraphs are so long that I think readers will get lost, and could consider additional sub-headings in the results. There is a TON of data here (RSV vs not, ARDS vs not, neutrophils, epithelial cells, etc). I just wonder if we were more strictly adhering to word limits if some of this could still be pared down and thus make the whole article easier for readers to follow & thus more impactful.

1. I'm still curious about the patient details. Was S1 truly captured in the first 48h? Table in supplement (pages 64-73+in the merged document) = what is the "time of sample"? IT looks like 160+ is that from admission? Are all of the S1 time points truly captured in the first day or 2?
2. In general, that table (spilling onto single columns) needs to be reformatted and legible for readers.
3. Median rather than mean for tables that are non-normal (time on vent, ages)
 - a. & similarly Wilcoxon rather than t test for significance for these features.
4. In general needs a lot more paragraphs—5 to 6 sentences max for readers to follow along. Whole pages of pure text are tough
5. There is a tremendous amount of data presented here. The discussion has a flow and interest, but even having read the paper 5x it's a struggle to keep following because of length. Could any of these analyses in the results be removed?
6. UMAPs would be more interpretable for readers if the scale were something other than 0-2. The descriptions also don't make it clear what the colors represent. Are these in some way P vs L vs C grouping together? If so, could that be in the scale instead of 0-2?

REVIEWER COMMENTS

Reviewer #1 (Remarks to the Author):

We appreciated the chance to review the revisions to Flerlage et al. In large part, we feel that the authors were very responsive to prior suggestions outlined in the initial review. In particular, we felt the figures were much clearer and easier to interpret than the original version and the inclusion of experimental limitations in the conclusions was helpful. We likewise appreciated the new analyses included in the manuscript. The quality of the dataset and accompanying analyses is high, and will undoubtedly provide an invaluable resource for specialists in a wide variety of fields. While enthusiasm for eventual publication is high, there remain some concerns about the presentation of the data.

Major point:

The key conclusions that the authors wish the reader to take away are obfuscated by the vast array of complementary findings that are simultaneously reported. As stated from their conclusions, “Our most salient findings are reduced interferon stimulation across multiple TA cell types, reduced proportional abundance of IL-10 expressing regulatory macrophages, and increased airway neutrophilia associated with a unique transcriptional profile in subjects with moderate to severe pARDS compared to subjects with no to mild pARDS.” While the richness of the dataset is fascinating, many of the reported results are not directly related to these key findings and ultimately distract from their appreciation. For example, both the virocap data and description of the FOLR3 finding (while interesting) are not essential for their key conclusions. These interesting auxiliary findings that are not central to their key results would be better suited for a separate manuscript or a supplemental results section.

The authors agree and feel that this is important feedback. In response, we have made several alterations to the manuscript that we feel addresses this point:

- We have changed the main figures of the manuscript to simplify the key messages we’re hoping to convey by removing additional analyses felt not to contribute to the message (for instance SCENIC analyses as well as Cell-Chat analyses (we have kept on figure from Cell Chat but moved it to supplementary material from a main figure (now figure S2d) to tie in this feedback with the feedback of reviewer 2 below).
- We have additionally reworded the key findings of this manuscript to include FOLR3 as a potential biological marker of interest to be investigated further in the future

Minor point:

--Please median line to violin plots. For some data, it is not clear which groups are higher than others.

We have responded to this in two ways – by adding median bars as well as by labelling violin plots with the actual median value for the group.

--More detailed titles and labeling of figure panels would be helpful (eg Fig S5).

We have added more detailed labelling to figure panels in the revised figures and feel that these changes address this minor point.

Reviewer #2 (Remarks to the Author):

Despite extensive revisions and redesigned analyses the manuscript is not substantially improved. It remains descriptive due to the small sample size, heterogeneity of patient groups by age, duration of illness, and lack of clarity regarding the infecting pathogens conclusions are unclear. Antecedent risk factors and treatments have not been carefully documented. The range of statistical/bioinformatics analysis is complex and poorly validated. Conclusions are not supported by orthogonal experimentation and are predominantly speculative. Because of the study design, lack of appropriate controls, and complexity of bioinformatic analyses, the conclusions drawn from the complex data set are limited.

We thank the reviewer for these remarks. Taking each in turn:

Small sample size- the study is currently closed to further patient enrolment and only open for analysis and manuscript preparation. Thus, we do not have the option to add more patients.

Heterogeneity of patient groups by age- There is heterogeneity of the patient groups by age, though all less than 2 years of age. We attempted to mitigate this initially by separating the “P” cohort into RSV pos and RSV neg subgroups for additional analysis and focused our comparisons on RSVposP v RSVposL, as there was no statistically significant difference in the age (in months) between the RSVposP and RSVposL subgroups. In further response to this point, we have plotted the absolute cell frequencies within each cluster within each cell population highlighted in the manuscript and stratified this by subject age, either above the median age for the entire cohort (3 months) or below the median age for the entire cohort. This analysis demonstrates that the cellular populations are similar between the entire dataset and the age less than 3 months. However, we have stressed this as a potential confounding factor in our analysis in the discussion section.

Duration of illness –this is a difficult variable to abstract because it is subjective, oftentimes not clearly remembered, and subject to recall bias. Thus, while it’d be nice to know which “day post infection” the subjects were at enrolment, this is not a variable we collected. We have noted this as a potential variable complicating the interpretation of our overall findings in the discussion section.

Lack of clarity regarding the infecting pathogens – we have included data regarding both clinical diagnosis for both viral and bacterial pathogens from routine clinical testing and have taken that one step further to include viral capture sequencing in tracheal aspirate supernatant to further confirm and supplement the clinical diagnosis in each case.

Antecedent risk factors and treatments have not been carefully documented – the study design in the methods lists the study inclusion and exclusion criteria for the study, which take care to enroll subjects who do not have known chronic cardiopulmonary conditions, immunologic, rheumatologic, or oncologic conditions and have not been prescribed corticosteroids prior to enrollment. We did not abstract treatments prior to hospitalization outside of corticosteroids. In all cases, though, this was the subject’s index hospitalization for this illness episode (i.e., they had not received care previously at another facility prior to being admitted to our institution).

Range of statistical/bioinformatics analysis is complex and poorly validated – In our initial submission, we had included analyses conducted using Cell Ranger and Seurat, which are commonly used analytic

pipelines for 10X single cell transcriptomic data. In response to reviewer's comments on the initial submission, we included SCENIC and Cell Chat analyses, which did not add significantly to the presentation of the data. Thus, these analyses have been removed in response to collective feedback from the reviewers on the revised manuscript, and we instead are focusing on analyses that are generated using Seurat (see below).

Conclusions are not supported by orthogonal validation and are predominantly speculative – we had attempted orthogonal validation in the revision. This was through measurement of analytes in a smaller sample size of tracheal aspirates collected at another institution. Herein, we confirmed FOLR3 is elevated in tracheal aspirates from subjects of higher disease severity in an independent cohort, which aligns with our data. Interestingly, we did also see a trend in decreased interferon alpha concentrations in higher illness severity subjects relative to lower illness severity subjects, which also aligns with our general findings.

We have taken the orthogonal validation question a step further in this revision and have stimulated monocytes obtained from otherwise healthy human donors (purified using magnetic assisted cell sorting from plateletpheresis rings) with diluted tracheal aspirate supernatants (leftover from our ViroCap studies) from RSVposP and RSVposL subjects. After an overnight incubation, we obtained RNA from adherent monocytes for PCR. With stimulation of monocytes followed by PCR of targets, we do observe differences in expression of several targets that align with single cell findings highlighted in our analyses of mononuclear phagocytes.

[figure redacted]

We do not have the option to do studies on cells as the study is closed, and we have concerns that any thawing of cryopreserved tracheal aspirate samples will lead to potential selective loss of our populations of interest (neutrophils and activated monocytes) as these are populations not typically amenable to cryopreservation and rethawing.

Specific concerns:

1) This is a diverse, heterogenous group of patients. Thus, the small study population remains a significant concern, as other reviewers expressed. This problem has not been addressed. The severe cohort was an average age of 5 months; the milder cohort was an average age of 1 month; the controls were ten months of age. The first year of life is marked by dynamic changes in immune maturation (see Olin et al., PMID: 30142345). An appropriate selection of age-matched cohorts is needed to support meaningful comparisons among the groups.

We agree that the first year of life is marked by dynamic changes in immune maturation and is an extremely important variable to consider. In the first revision, we addressed this, as discussed above, by separating the "P" cohort into "RSVneg" and "RSVposP" (based on whether RSV was detected). Comparing RSVposP to RSVposL (the lower illness severity group) demonstrates no statistically significant difference in the ages of these two subgroups. For this reason, most of our analyses and

presentation of findings focused on differences identified between RSVposP and RSVposL groups as these were groups matched based on infecting etiology without any statistical differences in age. We took this a step further in response to this feedback and have stratified the cohort based on age greater than or less than 3 months (which is the median age for the cohort as a whole). We then compared the absolute numbers of cells within each subclustered population (epithelial cells, mononuclear phagocytes, and neutrophils) between the cohort as a whole and the group of patients who were less than 3 months of age. We feel these data demonstrate that there is a similar composition of subclustered cell populations that support the analytic strategy we pursued.

[figure redacted]

2) A larger and as yet unresolved concern is that data linking severity with transcriptomic features are not supported by robust statistical analysis or mechanistic studies that validate the significant conclusions.

We have attempted orthogonal validation of single cell findings by stimulating healthy donor monocytes with diluted tracheal aspirate supernatant, the results of which are detailed above (Figure 1). We additionally measured analytes in multiplex in tracheal aspirates obtained at another hospital and found similar trends in this small sample, including most notably for FOLR3. As far as statistical analyses go, we are adding ROC analyses for gene sets we identified in this study as well as those we obtained from previously published literature (see below).

3) The new inclusion of analysis with CellChat and SCENIC provides little knowledge related to ARDS. For example, communication pathways associated with thromboinflammation (thrombospondin and c-type lectin)

were enriched in RSV pol L pathways (less severe). This seems counterintuitive as one would expect significant thrombosis and microvascular damage in clinically severe ARDS (reviewed in PMID: 27997925). Similarly, activation of IL1, cell apoptosis, and CCL signaling (pathways associated with antiviral responses) were more robust in L (less severe) group compared to the P (more severe groups), again seemingly counterintuitive.

The new inclusion of these analytic techniques was in response to reviewer commentary from the first submission. We agree that these did not significantly add to the overall story, and to address this comment as well as several others suggesting that we need to further simplify the presentation of data in text and figure formats, we have removed these additional analyses and have instead focused, as originally done, on analysis using Seurat.

Regarding the second point, while we agree that endothelial dysfunction and pulmonary microthrombosis likely occur in the context of ARDS, the cited review details several studies, many of

which focus on adults and base the framework of their discussion on the fact that ARDS results most commonly from sepsis or trauma (which are the most common causes of ARDS in adults).

For instance, here are the patients included one of the cited studies (Green R, et al. Pulmonary Vascular Obstruction in Severe ARDS: Angiographic Alterations after IV Fibrinolytic Therapy. *Am J Roentgenology* 148(3):465-653 (1987)):

[table redacted]

And another cited manuscript in the review (Dolinay T, et al. Inflammasome-regulated cytokines are critical mediators of acute lung injury. *AJRCCM* 185(11):1225-1234 (2012)), which performed comprehensive gene profiling in peripheral blood from patients with critical illness.

[table redacted]

In contrast, as mentioned in the introduction to the manuscript, lower respiratory tract infections are the most common cause of ARDS in children. Thus, we feel this critique speaks to the overall rationale for the study – that we know little about the airway side of the host response in infants to ARDS, as much of the literature focuses on old ARDS definitions in heterogeneous adult populations with different chronic comorbid conditions and underlying epidemiology of ARDS pathogenesis (direct versus indirect). Regarding the third point, this intuition would assume that increased apoptosis, interleukin 1 signaling, and ccl signaling would positively correlate with increasing degrees of illness severity in the context of viral pneumonia induced ARDS. This has not been shown and is another rationale for doing the study.

4) The authors have identified > 12 epithelial clusters, some of which represent canonical cell types. However, several clusters, for example, Epi_Int_RSV_1..... Epi_MT_High and Immune_HES likely represent transitory transcriptional states rather than distinct cell types and perhaps an artifact of over-clustering. It may be more informative to perform a pseudobulk RNA analysis with more robust DESEQ2-

based statistics (which the authors can do, given the multiple samples in each experimental group). Regardless of the analysis, the inference that epithelial responses to LRTI-associated pARDS depend on severity remains untested. This could be easily accomplished by identifying a 'unique transcriptomic signature in the epithelium in P and L cohorts and then calculating a ROC based on 'the defined transcriptomic signature' and disease severity as done by Wilk et al.

We agree that certain of annotated populations in the subclustered epithelial populations likely represent transcriptional cell states and perhaps not readily identifiable cell types, which is why we are careful in the manuscript to use transcriptional state and not cell population where appropriate. We have made sure that the language presented in the revision reflects this distinction. In response to this overall feedback, we have refocused the discussion of the epithelial cell populations to identifying general severity-based differences (module scoring, pathway analysis) as well as on RSV expression differences between groups, which we feel are appropriate analytic techniques for the number of cells we are including in the comparisons. We really appreciate the suggestion ROC to define a unique transcriptomic signature (as in Wilk et al) and have done so (Revised Figure 4g) and feel that this generally adds to the analysis of the epithelial cell populations.

5) The authors suggest that epithelial cell clusters expressing RSV genes are innate immune cells, which may have been co-captured in oil emulsion. While that certainly is possible, it is more likely that these represent contamination from more than one cell. How well were doublets accounted for in QI analysis; such a lineage switch has not been carefully validated is unprecedented in the literature.

We apologize for the confusion regarding this point. We are not speculating that there is a lineage switch from epithelial to immune cells (which I believe is what point 5 is suggesting is the takeaway from our discussion) as this is unprecedented and is certainly not our assertion. Rather, the possibility we suggest is that these are potentially reflective of “biological doublets”, meaning that we captured two cells interacting directly with one another through an immunologic synapse or other ligand-receptor mediated interaction in a bead within the oil emulsion. The alternative explanation would be that these are representative of technical doublets (i.e., cells that are stochastically co-captured in beads in the emulsion). As correctly pointed out, this discussion is purely speculative as we have no way of validating it, and we have taken care to ensure that this area of discussion is framed as a hypothesis to explain the presence of CD45 expressing cells in what we have subclustered as an “Epithelial” subpopulation. In the methods section, we had discussed our handling of doublets and poor-quality cells: “To exclude potential doublets, we removed cells that exhibited UMI counts or feature counts greater than the 99.5% probability quantile for each sample. Cells with greater than 20% of gene expression owing to mitochondrial genes were excluded as putatively dead or dying cells.”

6) Engagement of different inflammatory programs (Interferon/apoptosis vs. cytokine/immune cell accrual) by RSV- infected cells and bystander cells is expected and not novel. Notwithstanding the limited cell yields and the challenges of studying vulnerable populations, there is a need for orthogonal validation of major conclusions (FACS or ex vivo analysis). Furthermore, no experimental data support the concept that bystander cells coordinate immune responses and that bystander cells relieve intracellular stress.

We included these analyses (bystander and RSV-infected cells) at the recommendation of a reviewer following review of the first submission. We agree that these findings were not necessarily novel and are

similarly concerned about the power behind the findings given the fact that the analysis requires us to subset an already relatively small subsetting population for further comparison between subject groups. Given the concerns raised above and lack of ability to orthogonally validate findings given lack of access to samples with adequate cellularity and cell integrity, we have removed these analyses from the revision, which addresses this critique as well as those related to length and complexity raised by other reviewers.

7) Could the increased proportion of RSV-infected epithelial cells in the less severe groups be due to cell loss (dying cells not captured or discarded during QI)?

Yes, this could be the case, which is the rationale for using proportion of total cells as opposed to absolute cell numbers to try to mitigate some of these concerns. However, this remains a consideration, and we have added this as a potential limitation to the study. This is not unique to our study; however, and is a consideration for the interpretation of all single cell studies, especially those involving human specimens from infected compartments (like the airway).

8) Developmental trajectories directed from epithelial cell populations through the virus hi cluster to MNP clusters (Fig. 4i) is confusing and unprecedented. Do the authors imply that epithelial cells differentiate into MNP, or are the source of the epithelial cells as an 'inflammatory signaling wave' that propagates to MNP?

This implication of this analysis was that we were capturing differentiation trajectory from innate immune cells interacting with infected epithelial cells to what was seen in the mononuclear phagocytic cells. We in no way were suggesting that epithelial cells are differentiating into MNPs (as discussed further above). Given the confusion surrounding this point (as mentioned above), we have removed the trajectory analysis as it does not add significantly to the overall story and major take away points of the manuscript. Removal of this and its discussion further simplifies the manuscript in response to other reviewer's recommendation re: length and simplification.

9) Identifying distinct MNP types with pro-repair and inflammatory phenotypes could explain the differences in clinical severity. Unfortunately, the authors have not robustly linked transcriptional reprogramming of MNP with disease severity (an example as done by Wilk).

We appreciate this feedback and feel that we've linked transcriptional phenotypes to severity in similar manners to Wilk et al in JEM (i.e., we included module scoring using the gene lists published from their manuscript as well as gene lists we generated ourselves using our single cell data in this population) to show enrichment of individual transcripts as well as neutrophil and MNP gene expression modules within severity groups. We appreciate the suggestion above to include ROC analysis as Wilk et al had done in their manuscript and have included this in the revision (revised Figures 4g and 6f). Notably, our AUROC for these analyses are 0.70 ("P" epithelial), 0.78 ("L" epithelial), and 0.75 (aged neutrophil) at classifying illness severity amongst epithelial and neutrophil subpopulations, which are generally considered reflective of an informative classifier.

10) Accumulation of aged-hyperinflammatory neutrophils in P (severe) vs. L (less severe) groups, while novel in the context of pediatric LRTI, has been shown and validated orthogonally by FACS by several groups in human and animal models.

While we agree the concept of an aged neutrophil is not new (and we are not claiming this to be novel), we feel that finding their enrichment specifically within the airways of infants with severe RSV and describing their transcriptional phenotype in this context is an important finding for the field. Unfortunately, neutrophils notoriously do not do well with thawing following cryopreservation as is well known, and so we are unable to provide flow-based validation of these single cell findings.

Reviewer #3 (Remarks to the Author):

I do think the manuscript is improved vs the first round, but still suffers from being quite long and a little diffuse. The paragraphs are so long that I think readers will get lost, and could consider additional sub-headings in the results. There is a TON of data here (RSV vs not, ARDS vs not, neutrophils, epithelial cells, etc). I just wonder if we were more strictly adhering to word limits if some of this could still be pared down and thus make the whole article easier for readers to follow & thus more impactful.

The authors agree with this feedback and have addressed this major concern, which was voiced by another reviewer (reviewer 1). We have adjusted the main figures and text to reduce the amount of data presented so that the reader can focus on the salient points.

1. I'm still curious about the patient details. Was S1 truly captured in the first 48h? Table in supplement (pages 64-73+in the merged document) = what is the "time of sample"? IT looks like 160+ is that from admission? Are all of the S1 time points truly captured in the first day or 2?

Yes, sample 1 was truly captured in the first 48 hours (and in all cases but one, within the first 24 hours). The study design was to approach subjects for enrolment and collect the first endotracheal aspirate sample between 6- and 24-hours following intubation and initiation of invasive mechanical ventilation (not from admission). There was a question regarding appropriate group allocation (P vs L) for the subject whose sample was collected at 41 hours after intubation, which delayed enrolment after discussion with the IRB. We have adjusted the supplementary table, which details the timing of sample collection and other clinical variables abstracted from the EMR, to hopefully make it easier to follow.

2. In general, that table (spilling onto single columns) needs to be reformatted and legible for readers.

We agree with this feedback and have adjusted the table to hopefully improve its legibility for readers.

3. Median rather than mean for tables that are non-normal (time on vent, ages)

a. & similarly Wilcoxon rather than t test for significance for these features.

To our knowledge we did not include descriptive statistics in the supplementary tables. We utilized Kruskal Wallis testing with Dunn's post-hoc testing to compare individual groups to compare clinical variables that are presented in the main data figures. We have generated new figures that include the remainder of the clinical variables in graphical form in a supplementary figure to figure 1 with the appropriate statistical test as discussed above in this 3rd remark from reviewer 3.

4. In general needs a lot more paragraphs—5 to 6 sentences max for readers to follow along. Whole pages of pure text are tough

We agree with this feedback and feel it is an important point. In response, we have decreased the amount of data presented in both figure and text format and have separated long sections of text into smaller paragraphs.

5. There is a tremendous amount of data presented here. The discussion has a flow and interest, but even having read the paper 5x it's a struggle to keep following because of length. Could any of these analyses in the results be removed?

We feel that this is an important feedback point that was raised by other reviewers. We had initially responded to the first review of the manuscript by adding additional analyses (for instance using CellChat and SCENIC). Ultimately, those did not make a significant contribution to the general results of the manuscript and so to condense and simplify the presentation of data, we have removed entirely or relocated these analyses as appropriate to supplemental figures and adjusted the results section text accordingly.

6. UMAPs would be more interpretable for readers if the scale were something other than 0-2. The descriptions also don't make it clear what the colors represent. Are these in some way P vs L vs C grouping together? If so, could that be in the scale instead of 0-2?

We apologize for any confusion these figures have caused and have made effort to ensure that they are clearly labeled in the manuscript. As presented the UMAPs represent different variables throughout manuscript and are colored and labeled accordingly. As mentioned above, in the revision we have provided a lot more detail to annotation of the individual plots within each of the figure panels. The "UMAP 1 and 2" is to note that there are two dimensions being shown (these are standard outputs from the DimPlot command in Seurat, which was used to create all the UMAPs in the manuscript). We have not chosen to include "units" on these axes because these are arbitrary and do not add to the overall interpretation of the plots.

REVIEWERS' COMMENTS

Reviewer #1 (Remarks to the Author):

This manuscript has undergone extensive revisions and is substantially improved from the prior drafts. The manuscript now reads clearly and presents the extensive analyses in a logical manner. The authors' work represents a valuable addition to the growing knowledge base around pediatric acute respiratory distress syndrome, which is usually driven by a different etiology from its adult counterpart. The addition of external validation for the FOLR3 findings, the paring down of analyses that were otherwise underpowered, uninformative, or distracting, and the streamlining of the text to guide the reader through the extensive work performed here highlight the responsiveness of the authors to reviewer criticism and yield an article that merits publication. We would suggest only minor changes:

- Figure 1A does not match the description provided at the beginning of the section. 0-6 vs. 0-2 years of age. This is explained later, but could just be explained right at the beginning of the section describing subjects, prior to referencing the Figure.
- Figure 1C: axis label is cut off for the figure representing PICU time.
- Starting with the Figure 2, please consider consistency with group colors. C-Cohort is green for the first figure but is switched to black when the P-cohort is split. Please considering making C-Cohort black in the preceding figures.
- Barplots such as Figure 6C could use a more descriptive y-axis titles. A short description of the proportion being described would help the reader stay focused.

Reviewer #2 (Remarks to the Author):

- 1) The issue of small sample size is one of the critical handicaps of the manuscript as presented, one which unfortunately cannot be addressed as further enrollment in the study is closed. The approach used to mitigate the concerns of heterogeneity in the revision - which is further stratification of what is already, an underpowered dataset does not make sense.
- 2) Additional analyses, such as pseudo-time trajectory and cell-cell communication, clarified cell-type-specific transcriptomic remodeling nuances. Our concern in the initial round of review was related to the over-interpretation of data. Unfortunately, most of these analyses were removed from the resubmission, reducing this to 'just another scRNAseq study' and raising concern about the team's ability to interpret their own data and place it in the context of the current understanding of PARDS.
- 3) Limited analysis of transcriptome following stimulation of monocytes obtained from healthy adult donors with tracheal aspirates from the study subjects further obfuscates the study's conclusions. Is transcriptomic remodeling of MNP upstream of cytokine/chemokine/inflammatory protein secretion in the tracheal aspirates - OR - are the cytokines in the tracheal aspirates driving the transcriptomic changes in the MNP?
- 4) The validation cohort suffers from the same limitations as the primary cohort, is limited numerically, and is heterogenous in disease severity and pathogen exposure. It is not clear to us, how these additional data would transcend the weaknesses of the study.
- 5) While we recognize that this is an important area of research, we caution that an underdeveloped, poorly designed study is likely to obfuscate the current understanding of the role of developmental-stage and specific pathogen exposure in pediatric ARDS.

Reviewer #3 (Remarks to the Author):

This is an extensively revised manuscript by Flerlage et al based on single cell sequencing of 24 intubated pediatric patients, a majority of whom have RSV. The strengths of the manuscript remain the same (relative paucity of single cell data in pediatric ARDS, which is much rarer than adult, and thus relatively understudied, pulmonary samples, microbiome description for both viral and bacterial members, an additional focus on RSV vs not, and serial sampling over the course of illness in both the L and P groups.

The negatives remain that it is still a REALLY long article, and because it covers so much, a reader can still get lost in the major take-homes. In particular some of the details of viral and bacterial results could perhaps be moved to supplement to get more quickly to the focus on immune response between those with more severe vs mild disease, which strikes me as the most important take-home here.

Minor comment: Figure 4d and the discussion of “trends toward significant differences” in RSV counts doesn’t seem worth highlighting to the extent it is in the discussion (nearly a page), when the 4c data is clearly negative. Not sure this is worth focusing on much in the discussion at all, but just to say “not substantially different” and move back to the immune findings.

Reviewer #1 (Remarks to the Author):

This manuscript has undergone extensive revisions and is substantially improved from the prior drafts. The manuscript now reads clearly and presents the extensive analyses in a logical manner. The authors' work represents a valuable addition to the growing knowledge base around pediatric acute respiratory distress syndrome, which is usually driven by a different etiology from its adult counterpart. The addition of external validation for the FOLR3 findings, the paring down of analyses that were otherwise underpowered, uninformative, or distracting, and the streamlining of the text to guide the reader through the extensive work performed here highlight the responsiveness of the authors to reviewer criticism and yield an article that merits publication. We would suggest only minor changes:

- Figure 1A does not match the description provided at the beginning of the section. 0-6 vs. 0-2 years of age. This is explained later, but could just be explained right at the beginning of the section describing subjects, prior to referencing the Figure. **We thank the reviewer for this comment and we agree that this causes confusion in interpretation, we have moved the explanation to the beginning of the section prior to the figure reference.**
- Figure 1C: axis label is cut off for the figure representing PICU time. **Thank you for catching this, the plot position has been adjusted so that all axis labels are visible.**
- Starting with the Figure 2, please consider consistency with group colors. C-Cohort is green for the first figure but is switched to black when the P-cohort is split. Please considering making C-Cohort black in the preceding figures. **We thank the reviewer for this comment and we agree that this is a change that would reduce any confusion in interpretation – we have made the C-cohort black in the study design figures and clinical variables plots prior to splitting the P-cohort.**
- Barplots such as Figure 6C could use a more descriptive y-axis titles. A short description of the proportion being described would help the reader stay focused. **We thank the reviewer for this comment and we agree that this is a change that would reduce any confusion in interpretation of the bar plots, we have added a more complete description to the y-axis on the barplots.**

Reviewer #2 (Remarks to the Author):

1) The issue of small sample size is one of the critical handicaps of the manuscript as presented, one which unfortunately cannot be addressed as further enrollment in the study is closed. The approach used to mitigate the concerns of heterogeneity in the revision - which is further stratification of what is already, an underpowered dataset does not make sense. **We highlight the single center nature of this study being a limitation in the discussion section.**

2) Additional analyses, such as pseudo-time trajectory and cell-cell communication, clarified cell-type-specific transcriptomic remodeling nuances. Our concern in the initial round of review was related to the over-interpretation of data. Unfortunately, most of these analyses were removed from the resubmission, reducing this to 'just another scRNAseq study' and raising concern about the team's ability to interpret their own data and place it in the context of the current understanding of PARDS. **Other reviewers raised concerns about the length of the manuscript and we received feedback that the addition of these analyses did not add to the overall interpretation. Thus, we elected to remove these analyses to take everyone's critiques into consideration.**

3) Limited analysis of transcriptome following stimulation of monocytes obtained from healthy adult donors with tracheal aspirates from the study subjects further obfuscates the study's conclusions. Is transcriptomic remodeling of MNP upstream of cytokine/chemokine/inflammatory protein secretion in the tracheal aspirates - OR - are the cytokines in the tracheal aspirates driving the transcriptomic changes in the MNP? **This question is not answerable from the limited work we did in this area, but we**

are not presenting these data for publication.

4) The validation cohort suffers from the same limitations as the primary cohort, is limited numerically, and is heterogenous in disease severity and pathogen exposure. It is not clear to us, how these additional data would transcend the weaknesses of the study. The validation cohort provides confirmation of some key findings in the original cohort, despite the heterogeneity identified by the reviewer. As pARDS is a rare disease manifestation, we argue that it is important that these data are made available for the community and suggest that the accumulation of multiple distinct studies that repeatedly find shared correlates can help advance the field

5) While we recognize that this is an important area of research, we caution that an underdeveloped, poorly designed study is likely to obfuscate the current understanding of the role of developmental-stage and specific pathogen exposure in pediatric ARDS. We appreciate the reviewer would like us to have performed a study with more subjects over a longer period of time and have noted this as a limitation. To our knowledge, this still remains the largest study of primary pARDS samples examined at this level of resolution in the literature. We stand by the quality of the data that we have generated and have demonstrated using multiple, well-established metrics that the data are robust to much of the variation the reviewer is concerned about. We object to the depiction of our study design as poor as the reviewer has failed to raise any substantive issues with the quality of the results presented, but rather has raised disagreements of interpretation that are best sorted out in the broader literature.

Reviewer #3 (Remarks to the Author):

This is an extensively revised manuscript by Flerlage et al based on single cell sequencing of 24 intubated pediatric patients, a majority of whom have RSV. The strengths of the manuscript remain the same (relative paucity of single cell data in pediatric ARDS, which is much rarer than adult, and thus relatively understudied, pulmonary samples, microbiome description for both viral and bacterial members, an additional focus on RSV vs not, and serial sampling over the course of illness in both the L and P groups.

The negatives remain that it is still a REALLY long article, and because it covers so much, a reader can still get lost in the major take-homes. In particular some of the details of viral and bacterial results could perhaps be moved to supplement to get more quickly to the focus on immune response between those with more severe vs mild disease, which strikes me as the most important take-home here.

Minor comment: Figure 4d and the discussion of “trends toward significant differences” in RSV counts doesn’t seem worth highlighting to the extent it is in the discussion (nearly a page), when the 4c data is clearly negative. Not sure this is worth focusing on much in the discussion at all, but just to say “not substantially different” and move back to the immune findings. We thank the reviewer for this critique and agree. In response, we have shortened the portion of discussion related to differences in RSV gene expression.